# Towards a more effective climate policy on international trade

Erik Dietzenbacher [1✉], Ignacio Cazcarro [2,3] & Iñaki Arto [3]

In the literature on the attribution of responsibilities for greenhouse gas emissions, two accounting methods have been widely discussed: production-based accounting (PBA) and consumption-based accounting (CBA). It has been argued that an accounting framework for attributing responsibilities should credit actions contributing to reduce global emissions and should penalize actions increasing them. Neither PBA nor CBA satisfy this principle. Adapting classical Ricardian trade theory, we consider ex post measurement and propose a scheme for assigning credits and penalties. Their size is determined by how much $CO_2$ emissions are saved globally due to trade. This leads to the emission responsibility allotment (ERA) for assigning responsibilities. We illustrate the differences between ERA and PBA and CBA by comparing their results for 41 countries and regions between 1995–2009. The Paris Agreement (COP21) proposed new market mechanisms; we argue that ERA is well suited to measure and evaluate their overall mitigation impact.

[1] University of Groningen, Faculty of Economics and Business, PO Box 800, 9700 AV Groningen, Netherlands. [2] ARAID (Aragonese Agency for Research and Development), Agrifood Institute of Aragon (IA2), Department of Economic Analysis. Faculty of Economics & Business Studies, University of Zaragoza, Zaragoza 50005, Spain. [3] Basque Centre for Climate Change, Scientific Park of the University of the Basque Country (UPV/EHU), Edificio Sede 1, Planta 1ª | Parque Científico de UPV/EHU, 48940 Leioa (Bizkaia), Spain. ✉email: h.w.a.dietzenbacher@rug.nl

Two main accounting methods are used in the literature to determine countries' contribution to global emissions or environmental problems. The production-based accounting (PBA) measures the amount of, e.g., $CO_2$ released to the atmosphere by the industries and households of a country. The consumption-based accounting (CBA) attributes emissions to the country's consumption of final products. CBA redistributes the emissions from PBA and considers that emissions in another country are necessary for the home country's consumption bundle. Foreign emissions are embodied in the intermediate inputs necessary to produce the home country's consumption goods and in the final goods directly imported by the home country's consumers. The home country is said to import foreign emissions.

PBA and CBA are used to assign responsibility for global environmental problems, such as climate change to countries. Producer responsibility addresses the countries that directly generate the pressure based on PBA. Consumer responsibility addresses the countries that ultimately drive the pressure. Another possibility is to share producer and consumer responsibility[1–3]. Also the historical responsibility has emerged in climate negotiations, leaving developing countries better positioned in negotiations if the time frame is shifted backward to the pre-industrial time. Theoretically speaking, historical trade-adjusted emissions accounts could be developed. In practice, however, there are enormous data challenges and methodological issues. Trade-adjusted accounts have been made only for the last three decades. An increasing number of authors have examined the nexus of producer–consumer responsibility, often dealing with how to assign responsibility for internationally traded greenhouse gas (GHG) emissions.

It seems an obvious step to use PBA and CBA for designing a scheme of credits and penalties to guide policy actions. Kander et al.[4] rightfully stated that "actions that contribute to reduced global emissions should be credited, and actions that increase them should be penalized." However, neither PBA nor CBA satisfy this principle. The discussion on carbon leakage and pollution havens shows that PBA cannot be used for valuing countries' contributions to global emission reduction. A country may decide to import a product from abroad where it is produced more emission-intensively than at home. In this case, global emissions increase but the home country is rewarded because its PBA decreases. CBA fails to credit countries for cleaning up their export industries[4]. In this case, global emissions decrease but the country is not rewarded because its CBA does not decrease.

Kander et al.[4] adapt CBA and propose technology-adjusted CBA (TCBA) to remedy this weakness and Domingos et al.[5] propose TCBA∗ as a further adaptation. However, as the example in Supplementary Note 2 of the Supplementary Information (SI) shows, also TCBA and TCBA∗ may penalize a country to engage in trade that reduces global emissions (even to a larger extent than does CBA). The reason is that they do not fully account for the second weakness of CBA mentioned by Kander et al.[4]: "CBA fails to encourage certain kinds of specialization and trade that might contribute to a more carbon-efficient use of global production resources".

We argue that ex post emission accounting is one thing, but providing ex ante incentives to engage in a certain type of trade is another. We link classical Ricardian trade theory to environmental concerns, yielding a variant where countries should export what they produce most clean. Standard emission multipliers indicate ex ante where consumers should buy their products to reduce global emissions. Using this information, ex post accounting indicates how a country performs compared to other countries (or the world average). This yields a scheme of credits and penalties. A credit quantifies whether (and to what extent) a country has reduced—through its trade—global emissions more than the average country did. Adapting CBA with the credits and penalties results in emission responsibility allotments (ERAs).

The size of the credits and penalties is determined by $CO_2$ emissions saved globally due to the trade of this country (when compared to the savings achieved by the average country). They satisfy the properties of sensitivity (i.e., credits and penalties are responsive to factors that a country can influence), additivity (i.e., global emissions are the sum of all national emissions), and monotonicity (i.e., if the actions of country A reduce global emissions more than average and more than the actions of country B do, country A should be credited and more than country B). These three properties were identified as desired properties by Kander et al.[4] and included in the larger set of desired properties in Rodrigues et al.[6] and Domingos et al.[5]. In summary, PBA and CBA do not meet the properties described in the above literature, nor do TCBA and TCBA∗ (which adapt CBA). We also adapt CBA with credits and penalties, and propose ERA as a simple method to account for responsibilities.

ERA can be a powerful tool to track and assess how successful countries can reduce global emissions through changing their production, consumption or trade patterns. Several voices stressed the relevance of having key indicators to track the progress of individual and collective contributions in the context of the Paris Agreement (COP21)[7–10]. ERA can serve to track joint actions comprehensively and over time. This information can be useful in periodical global stocktaking[11]. Main sources of information to evaluate this progress are the national emissions inventories, which follow PBA. However, as indicated in the COP21, any comprehensive assessment should look into the global effects of reductions of emissions. Moreover, the Agreement (article 6) also provided the opportunity to use internationally transferred mitigation outcomes (ITMOs). These are thought to be implemented mainly bilaterally, where country A pays money to country B for the emission reductions carried out in B and attributed to A. The ITMOs require a robust accounting system and should be guided by the fundamental principles of ensuring environmental integrity and avoiding double counting[12,13], which align with the ERA approach.

Applying our method to the World Input–Output Database (WIOD)[14], we find that the ERAs are very close to the CBA outcomes. This implies that the credits and penalties are generally small, which means that most countries perform very similar when it comes to reducing global emissions through trade. Nevertheless, if ERA were adopted, rich countries would on average receive credits for their trade and poor countries pay penalties. The most remarkable outcomes for separate countries are the large penalties for the USA and China, which is primarily caused by trade between these two countries.

## Results

**Application to the WIOD.** In this section, we present the results for ERA (and compare them with those for other methods). The discussion section comments on the method, results, and the opportunities to use ERA for climate change policies. For the empirical application, we use WIOD[14] and the corresponding emissions data from the environmental satellite accounts[15]. These accounts give the $CO_2$ emissions from human sources (i.e., not all anthropogenic $CO_2$). The main text focuses on $CO_2$ emissions (including $CO_2$ emitted directly by households), the SI provides additional results for $CO_2$, $CH_4$, and $N_2O$ in $CO_2$ equivalents.

The environmental accounts in WIOD are consistent with the input–output (IO) data. That is, they apply the residence principle where emissions of a resident (no matter whether she is physically present in the territory) are allocated to the territory

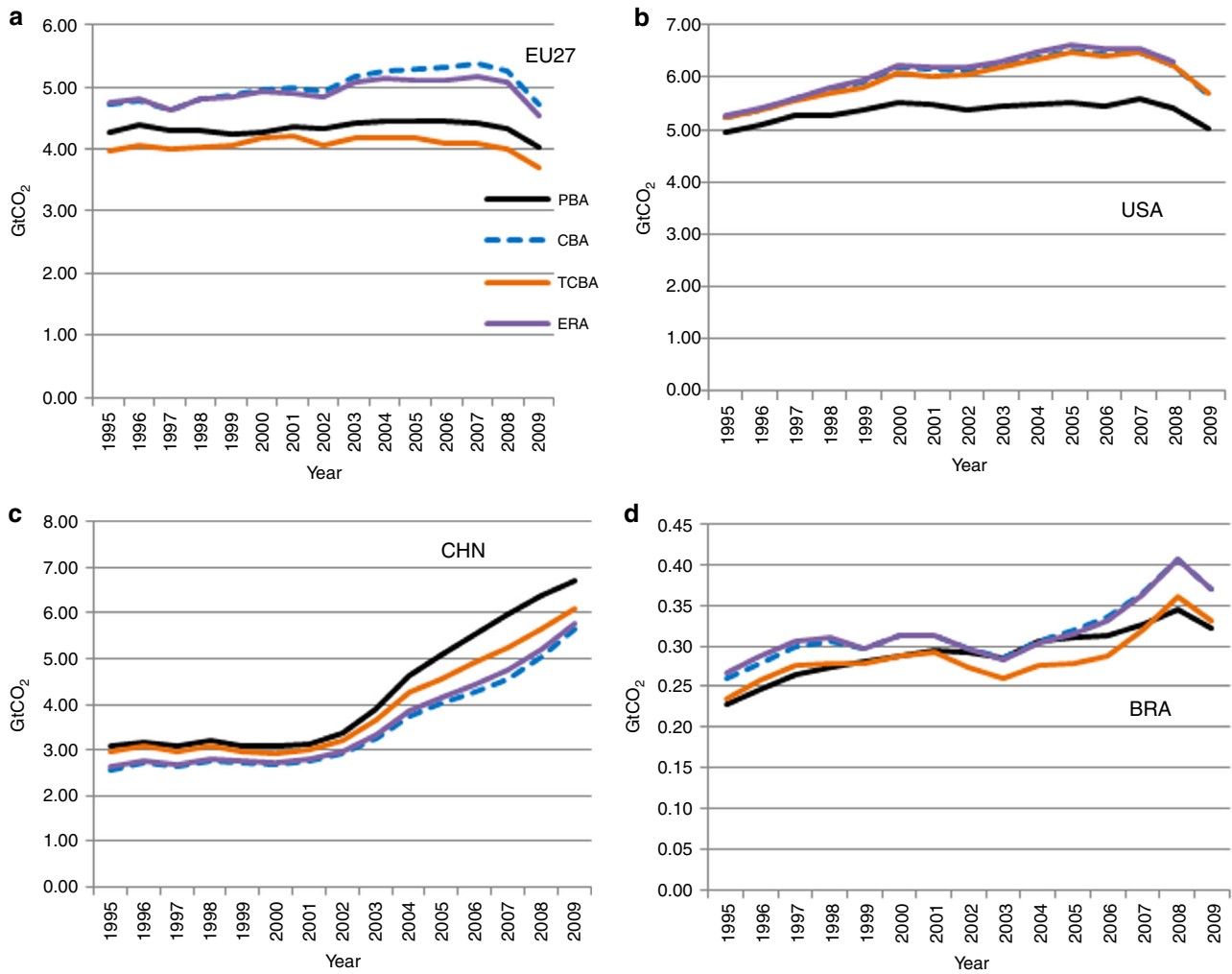

**Fig. 1 A comparison of emission accounting methods.** Each graph gives the results when applying production-based accounting (PBA, black), consumption-based accounting (CBA, dashed blue), technology-adjusted CBA (TCBA, orange), and emission responsibility allotment (ERA, purple). The graphs are given for the EU27 **a**, the USA **b**, China **c**, and Brazil **d**, and give GHG emissions in Gt $CO_2$ equivalents.

of residence. This is different from the territorial principle underlying energy balances and emissions inventories. The territorial principle allocates emissions to the country in which they physically take place, regardless of whether they are undertaken by residents or non-residents[15]. There are 41 countries (including the Rest of the World, RoW) in WIOD, each with 35 industries. The countries are: 27 countries of the EU, Australia (AUS), Brazil (BRA), Canada (CAN), China (CHN), India (IND), Indonesia (IDN), Japan (JPN), Korea (KOR), Mexico (MEX), Russia (RUS), Taiwan (TWN), Turkey (TUR), the USA, and RoW.

In Fig. 1, we compare ERA with PBA, CBA, and TCBA (all for $CO_2$ emissions) for the EU27, the USA, China, and Brazil. Supplementary Table 1 in the SI presents the results for PBA, CBA, TCBA, TCBA∗ (which corrects the TCBA)[5,16], and ERA for 1995–2009 for all 41 countries individually. Supplementary Table 2 does so for emissions of $CO_2$, $CH_4$, and $N_2O$.

The most striking finding is that the ERAs are very close to the CBA. This means that the credits or penalties of countries are in general very small. A small credit for country $R$ means that its trade relations reduce global emissions only slightly more than do trade relations on average. This implies that, all in all, countries behave rather similar and there are no clear heroes and antiheroes in terms of trade that reduces global emissions.

Of course, some countries will certainly be heroes (and some other countries anti-heroes) of cleaner production. This could be the case, for example, because they have installed technologies with low emission intensities. The countries are then rewarded for their actions by lower PBAs and CBAs. ERAs adapt CBAs by looking at extra credits that can be gained only if a country trades the appropriate goods with the right partner. However, the questions what to trade and with whom to trade are in real economic life still answered on the basis of profitability and not of environmental considerations. This implies that few credits are gained from trade, which may explain why we find ERAs so close to the CBA. At the same time, if countries do not use environmental considerations when deciding on trade, there is a huge opportunity to gain many credits for the first countries that decide to follow the principles of what might be termed green Ricardian trade.

**Credits and penalties.** Let the scalar $ERA^R$ ($CBA^R$) denote the ERA (CBA) of country $R$, then the credits/penalties are obtained from $ERA^R - CBA^R$. A negative difference ($ERA^R - CBA^R < 0$) indicates a credit and a positive difference a penalty. Note that the sum of credits and penalties equals zero, i.e., $\sum_R (ERA^R - CBA^R) = 0$. This implies that the credits/penalties

**Table 1 Differences between ERA and CBA (in Mt $CO_2$, 2009).**

| 20 Richest countries* (by GDP pc) | $ERA^R - CBA^R$ | As % of $CBA^R$ | 20 Poorest countries* (by GDP pc) | $ERA^R - CBA^R$ | As % of $CBA^R$ |
|---|---|---|---|---|---|
| LUX | −3.68 | −46% | CZE | −3.66 | −4% |
| USA | 71.58 | 1% | KOR | −4.35 | −1% |
| IRL | −9.44 | −16% | MLT | −3.30 | −93% |
| NLD | −2.89 | −1% | PRT | −4.19 | −6% |
| CAN | 6.38 | 1% | SVN | −3.90 | −19% |
| AUT | −4.96 | −5% | EST | −3.84 | −31% |
| SWE | −6.80 | −8% | HUN | −3.92 | −7% |
| AUS | 3.51 | 1% | LVA | −3.13 | −29% |
| BEL | −2.97 | −2% | LTU | −3.49 | −18% |
| DNK | −8.35 | −13% | POL | −3.04 | −1% |
| FIN | −5.29 | −8% | RUS | −49.15 | −4% |
| GBR | −5.48 | −1% | MEX | 2.82 | 1% |
| GER | −31.05 | −3% | BGR | −4.00 | −11% |
| SPA | −1.65 | 0% | ROM | −3.58 | −4% |
| JPN | 3.61 | 0% | TUR | 1.02 | 0% |
| FRA | −8.73 | −2% | RoW | −30.53 | −1% |
| GRC | −9.24 | −7% | BRA | −0.37 | 0% |
| TWN | 0.23 | 0% | CHN | 129.17 | 2% |
| ITA | −16.62 | −3% | IDN | 2.06 | 1% |
| SVK | −3.34 | −9% | IND | 28.53 | 2% |
| CYP | −3.96 | −35% | | | |
| Total richest 20 | −35.18 | −0.3% | Total poorest 20 | 39.14 | 0.2% |
| Total except USA | −106.76 | −1.6% | Total except China | −90.02 | −0.8% |

*Both columns rank the countries from richest to poorest. Credits are negative and penalties are positive, the USA and China have the largest penalties. CYP has the median GDP per capita and is therefore excluded from both top 20 lists (richest or poorest).

may be blown up and multiplied by a factor 2, 5, 10, or any number, based on policy decisions. Therefore, we stress the difference between an accounting framework (like CBA or PBA, which are positive in nature) and a scheme of credits/penalties (like ERA, which is normative, but based on positive information about the reduction in global emissions). Although differences between $ERA^R$ and $CBA^R$ are very small in Fig. 1, their sign is still relevant because it remains the same if credits/penalties are blown up. The results show that for the EU27 ERA lies below CBA (implying small credits), while for the USA and China ERA lies above CBA (implying penalties).

Next, we consider (for each country) $ERA^R - CBA^R$ in the year 2009. Recall that a negative outcome implies a credit for this country and a positive outcome implies a penalty. Countries are ranked according to gross domestic product (GDP) per capita and split into the 20 richest regions and the 20 poorest regions (Cyprus ranks at place 21 and has the median GDP per capita).

The results in Table 1 show that the richest countries (with a total credit of 35 megatonnes of $CO_2$, Mt $CO_2$) trade better (i.e., in general with higher credits) than the poorest countries (with a penalty of 39 Mt). It is also clear that the results are dominated by the very large penalties for China (129 Mt $CO_2$) and the USA (72 Mt $CO_2$). Consequently, most remaining rich countries (all 20 except the USA) still trade better than most remaining poor (all 20 except China), but the gap between rich and poor has nearly vanished (total credits of 107 and 90 Mt $CO_2$), and almost all perform better than the average. Countries with large credits are Russia (49 Mt $CO_2$), Germany (31), RoW (31), and Italy (17); a large penalty is for India (29 Mt $CO_2$).

Clearly, these results partly reflect size. In terms of a country's achievements, it may be better to consider the relative credits and penalties (as percentage of the CBA). Small countries trade very well: Luxemburg (with a credit that is 46% of its CBA), Cyprus (35%), Ireland (16%), and Denmark (13%). The EU27 consists of many small countries that trade a lot with each other. The group

**Table 2 Summary of credits and penalties in 1995 and 2009 (in Mt $CO_2$).**

| | 1995 | | 2009 | |
|---|---|---|---|---|
| | Rich | Poor | Rich | Poor |
| Credit ($ERA^R - CBA^R < 0$) | 9 | 3 | 15 | 15 |
| Penalty ($ERA^R - CBA^R > 0$) | 11 | 17 | 5 | 5 |

of 22 smaller EU countries (EU27, except the larger countries Germany, France, UK, Italy, and Spain) has a credit of 101 Mt of $CO_2$ (which is 7% of its CBA) and 37 Mt is due to trade within the group.

The most remarkable finding are the large penalties for the USA and China. Supplementary Table 3 in the SI gives the credits and penalties (i.e., ERA minus CBA) at the bilateral level. By far the largest outcome in size is the penalty of 61 Mt of $CO_2$ for the trade between the USA and China. The major contributor to this penalty is electrical and optical equipment (industry c14). Trade between China and the USA regarding the final products of this industry causes 36 Mt of the penalty. Other industries that contributed substantially are textiles and textile products (6 Mt $CO_2$), machinery Nec (5 Mt $CO_2$), and manufacturing Nec (4 Mt $CO_2$). In 1995–2009, trade among the two countries grew more than six times on a nominal basis. A trade war between these countries, where imports from each other are not allowed, would have at least been beneficial for the global $CO_2$ emissions in 2009.

The differences between 1995 and 2009 are summarized in Table 2. In 1995, most of the countries (28 out of 40) had a penalty and more poor countries than rich countries (according to the 2009 GDP per capita) had a penalty (and thus more rich than poor countries a credit). The situation in 2009 was reversed

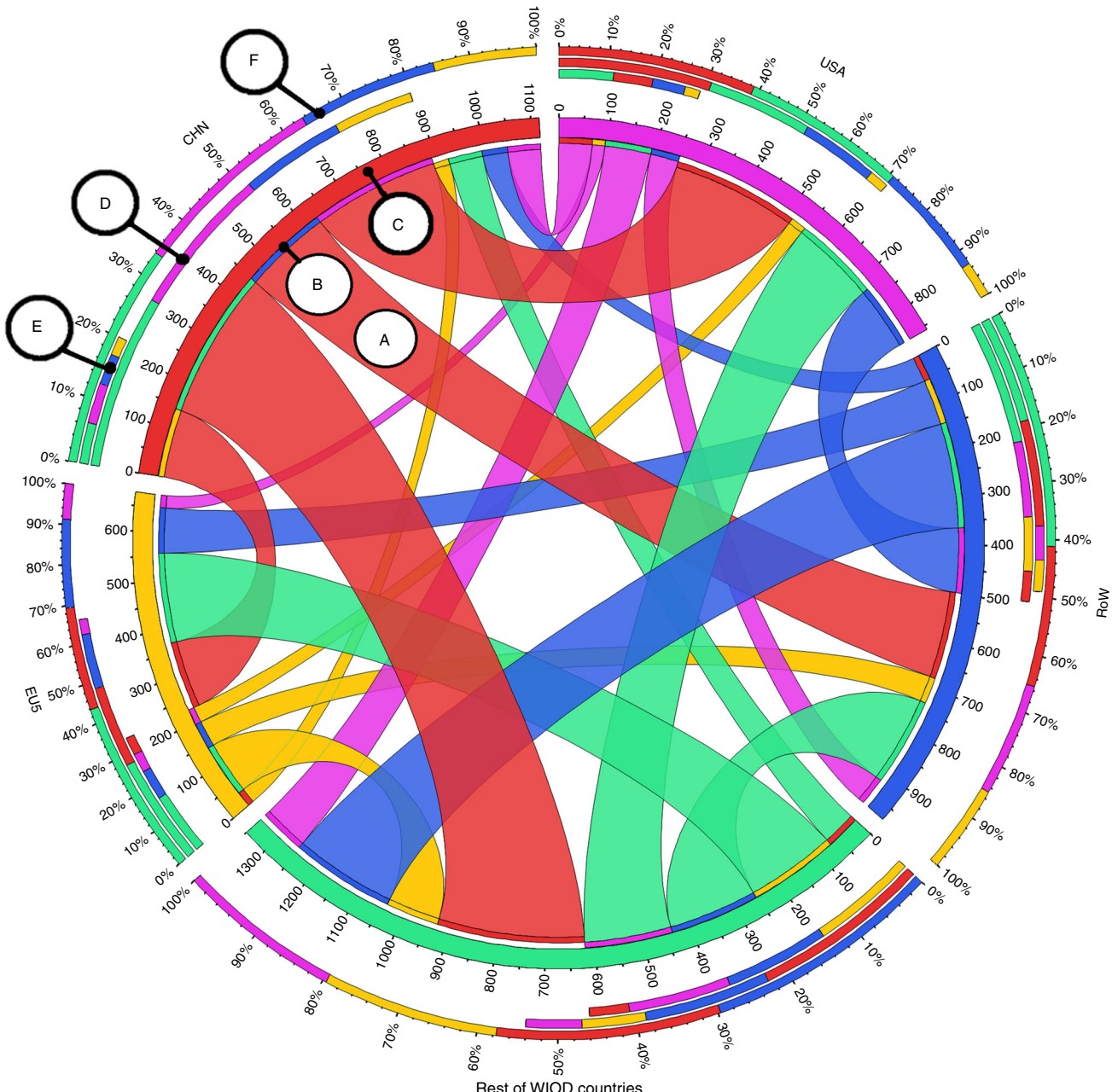

**Fig. 2 Circos graph with CO$_2$ ERAs (in Mt CO$_2$) by origin and destination.** This graph distinguishes five regions as origin and destination. These are: USA (the pink country); China (the red country); the EU15 (the yellow region); the Rest of WIOD countries (aggregate of 22 countries in WIOD, the green region); and the rest of the world (RoW) in WIOD (the blue region). The explanation is given by the items A–F. A: Export of emissions (in Mt CO$_2$) from China (the red country) to RoW (the blue region), after correcting for the penalty that China and RoW receive because their bilateral trade contributes less to global emission reduction than the average bilateral trade. B: The inner most ribbon gives the color of the importing country, for example blue because RoW is the destination. C: The second ribbon gives the color of the exporting country. D: The third ribbon also gives the exports of emissions (in Mt CO$_2$) split according to destination countries/regions. E: The fourth ribbon gives the imports (in Mt CO$_2$) split according to origin. F: The outer ribbon gives the share of the exports that each destination receives.

(more credits than penalties) and balanced (the numbers of rich and poor countries receiving a credit—or penalty—were the same).

**Trade in emissions**. Figure 2 visualizes the trade in emissions in terms of ERAs, which allows to examine who emits for whom. It gives the exports of emissions (following the CBA approach at the bilateral level) that is corrected with the penalty or credit for each pair of countries. Because the differences between ERA and CBA are fairly small (as follows from Supplementary Table 3 in the SI), the circos graph with CBA-based exports and import of emissions looks very similar. The center of the graph gives the flows, the five ribbons around the center provide the same information in a different way (as explained in the note to Fig. 2). In order to be able to focus on the flows that really matter, we aggregate the EU5 (Germany, UK, Italy, France, and Spain), and the non-presented countries from WIOD in the RoW.

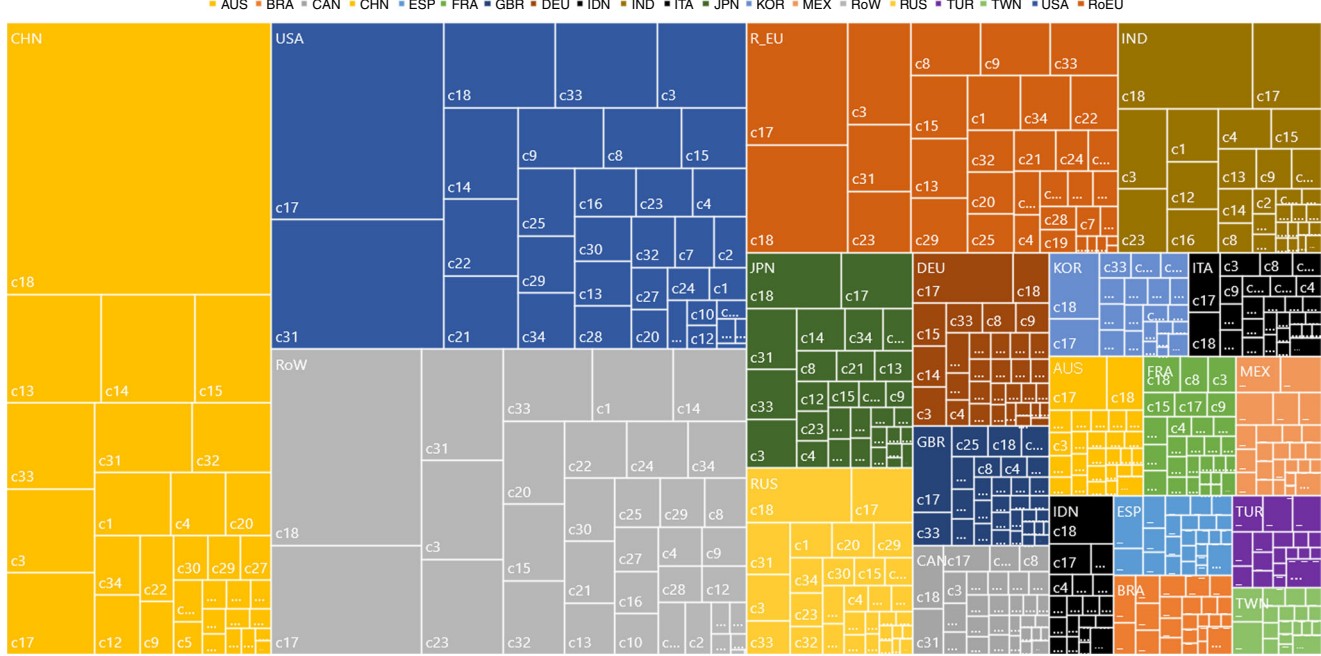

**Fig. 3 Tree map of ERAs in 2009 by selected country and industry.** Each rectangle reflects the size of the ERA in a particular industry in a certain country. For example, the upper left rectangle gives the ERA for the construction industry in China. Going from top to bottom and from left to right, the ERAs become smaller. The countries are: Australia (AUS); Brazil (BRA); Canada (CAN); China (CHN); France (FRA); Great Britain (GBR); Germany (DEU); Indonesia (IDN); India (IND); Italy (ITA); Japan (JPN); Korea (KOR); Mexico (MEX); Rest of the EU (RoEU, an aggregate of the 22 remaining EU countries in WIOD); Rest of the World in WIOD (RoW); Russia (RUS); Spain (ESP); Turkey (TUR); Taiwan (TWN); and the USA. The industries are classified as follows. c1: agriculture, hunting, forestry, and fishing; c2: mining and quarrying; c3: food, beverages, and tobacco; c4: textiles and textile products; c5: leather, leather, and footwear; c6: wood, and products of wood and cork; c7: pulp, paper, printing, and publishing; c8: coke, refined petroleum, and nuclear fuel; c9: chemicals and chemical products; c10: rubber and plastics; c11: other non-metallic mineral; c12: basic metals and fabricated metal; c13: machinery, Nec.; c14: electrical and optical equipment; c15: transport equipment; c16: manufacturing, Nec.; recycling; c17: electricity, gas, and water supply; c18: construction; c19: sale, maintenance, and repair of motor vehicles and motorcycles; retail sale of fuel; c20: wholesale trade and commission trade, except of motor vehicles and motorcycles; c21: retail trade, except of motor vehicles and motorcycles; repair of household goods; c22: hotels and restaurants; c23: inland transport; c24: water transport; c25: air transport; c26: other supporting and auxiliary transport activities; activities of travel agencies; c27: post and telecommunications; c28: financial intermediation; c29: real estate activities; c30: renting of M&Eq. and other business activities; c31: public admin and defence; compulsory social security; c32: education; c33: health and social work; c34: other community, social, and personal services; c35: private households with employed persons.

The flows have the color of the exporting country. A large flow is 182 Mt $CO_2$ for the exports of Chinese emissions (in red) to RoW. This includes the 18 Mt penalty for trade between China and RoW. Other large flows involving single countries are: from China to USA (263 Mt $CO_2$), from China to EU5 (141 Mt $CO_2$), and from RoW to USA (133 Mt $CO_2$, colored in blue). Clearly, China is a net exporter, and the USA and EU5 net importers. Supplementary Fig. 1 in the SI gives the graph for a larger number of countries, and Supplementary Fig. 2 also for emissions of $CO_2$, $CH_4$, and $N_2O$.

**Countries and industries**. Figure 3 is a tree map of ERA results by selected countries and industries in 2009. Countries are ordered from left to right and from top to bottom according to their ERAs. Each country has a different color and the size of the rectangle for country $R$ reflects $ERA^R$. Within each country, the contributions by each industry are ordered from Northwest to Southeast according to the contribution size ($ERA_i^R$ as given in subsection 1.5 of Supplementary Note 1 of the SI). It measures the global emissions embodied in country $R$'s consumption of final product $i$ (irrespective of the origin) corrected with a credit/penalty for country $R$'s trade in product $i$. In Fig. 3, we present the five largest economies in the EU separately (France, Germany, Italy, Spain, and the UK) and aggregated the results for the remaining 22 EU countries as Rest of the EU (RoEU).

The role of separate countries and groups of countries becomes immediately clear. China, the USA, and the RoW are of comparable size and together responsible for 60% of global emissions. China (20%) and the USA (18%) are the largest players, and contribute each as much as the ~150 (largely developing) countries aggregated in the RoW (21%). The industries that contribute the most are similar across countries. These are: construction (c18, first for China, RoW, and India, third for USA); electricity, gas, and water supply (c17, first for USA and the main EU countries, second for RoW and India, while of lesser importance for China), public admin and defence (c31, second for USA and RoW, sixth for China, while of relatively low importance in India), health and social work (c33, fourth for USA, fifth for China, and sixth for RoW), and food, beverages, and tobacco (c3, third for India, fourth for RoW, fifth for USA, and eighth for China).

Supplementary Table 4 in the SI gives the major differences between ERA and CBA in 2009. A positive difference indicates a penalty. It means that trade of final product $i$ by country $R$ increased global emissions more (or reduced them less) than trade of final product $i$ increased (decreased) global emissions on average. China's trade for the industry electrical and optical equipment (c14, which ranked third in total ERA value, see Fig. 3) led to an enormous penalty of 68 Mt $CO_2$ (about half of the country's overall penalty of 129 Mt $CO_2$). It indicates that China's

focus on the exports of final products from c14 was environmentally rather damaging.

An important penalty (37 Mt $CO_2$) is also found for the same sector for the USA, followed for this country by manufacturing Nec (c16, a penalty of 13 Mt $CO_2$) which is not a sector with a particularly high absolute value of ERA for the USA. Other Chinese industries with large outcomes are: textiles products (c4, penalty of 28 Mt $CO_2$), basic and fabricated metals (c12, penalty of 7 Mt $CO_2$), and manufacturing, Nec (c16, penalty of 6 Mt $CO_2$). Interestingly, none of these accounts are among the highest ERAs for China. On the contrary, the clear-cut highest ERAs were for construction (c18) and machinery, Nec (c13), which had a penalty and a credit of 1 Mt $CO_2$, respectively. For other countries, high penalties are found also for the sector manufacturing, Nec (c16) in India (penalty of 14 Mt $CO_2$) and the RoW (penalty of 12 Mt $CO_2$), while an important credit (of 29 Mt $CO_2$) for the RoW in water transport (c24).

These findings highlight the importance of industry detail for this type of analyses and for policy implementation. It is relevant to know which industries export many final products (but the framework can be extended to intermediate products) and how many global emissions they embody. This information may be used as a guide in developing a trade strategy that reduces global emissions. Of course, also other considerations matter when deciding about trade. For example, full allocation of responsibilities to consumers may imply financial problems in some industries[16]. For certain questions and topics also regional detail is of crucial importance. For instance, quantifying the effects of trade among developing nations[17].

Further visual analyses on deviations across countries/sectors/time, and sensitivity analyses illustrating possible uncertainties involved are shown in Supplementary Note 5.

## Discussion

In recent years, global multi-regional IO (GMRIO) tables and corresponding IO models are increasingly used to account emissions and their trade, helping to design or improve climate policy. Examples include accounting carbon footprints[18,19], computing balances of avoided emissions[20,21], or finding the drivers of changes in global GHG emissions[22].

Starting from emission multipliers, we adapted Ricardian trade theory. Countries specialize in and export the goods and services that they produce relatively the cleanest. This allowed us to answer ex post the question how much global emissions each country reduced through its trade, which was translated into a scheme of credits and penalties. Adapting the responsibilities from the CBA approach with the credits and penalties resulted in the ERAs. The scheme of credits and penalties satisfies the criteria of sensitivity, monotonicity, and additivity (three desired properties)[4]. Earlier adaptations of CBA (i.e., TCBA[4] and TCBA*[5]) did not meet the condition of monotonicity.

In past agreements on emissions and climate policies, the role of credits/penalties was central. The Kyoto Protocol provided three mechanisms meant to help countries control their emissions through flexible arrangements[23]. First, the Clean Development Mechanism allowed industrialized countries to invest in climate-friendly projects in poor countries for which they earned carbon credits (Certified Emission Reductions). Second, the Joint Implementation mechanism enabled industrialized countries to invest in climate-friendly projects in other industrialized countries for which they received another type of carbon credits (Emission Reduction Units). Third, the trade in emissions created a market for trading carbon credits. For example, countries that passed their target could buy carbon credits. This led to pricing systems, such as cap-and-trade in the EU emissions trading system[23].

It is likely that credits and penalties will remain important also in the future. The COP21 redefined the framework for designing climate policies. The agreement has been analyzed thoroughly and the studies on the implications are numerous and diverse[24–29]. Some clear messages were given and climate proposals need a boost to keep warming well <2 °C (ref. [30]). The COP21 also concluded to replace existing market mechanisms by a new one after 2020. Article 6 of the agreement assumes that countries will develop internal markets and it says that countries can trade ITMOs to reach the targets they have set in their nationally determined contributions (NDCs).

Many scientists, observers, and governments have been calling for some time for this idea of countries working together to reduce emissions. ITMOs should be designed to deliver an overall mitigation impact. That is, a net mitigation impact rather than purely an offsetting mechanism as conceived under the Kyoto Protocol. International trading of certificates or ITMOs is only possible when the rules for accounting the achieved emission reductions are clearly agreed upon and double counting of certificates is avoided[31]. This requires a robust accounting system that is generally agreed upon. Typically, arriving at such an agreement is a challenge. However, when the consumer perspective is adopted, most of the methodical issues have been solved[32,33], which leaves the choice of a database and the aggregation level to agree upon. ERAs use the same type of information and methodology as the CBA approach. ERAs are thus well suited to evaluate the ITMOs in terms of global mitigation impact. Also, political leaders have agreed to a global, net decarbonization of human activities before 2100. For this and for other objectives, ERAs may play an important role in the evaluation of the expected and real trajectories of national and global emissions.

PBA and CBA will remain very valuable tools to assess these trajectories from the producer and consumer perspective. PBA provides the information that is used as an input for CBA, TCBA, and ERA. Nevertheless, more researchers and policy makers are looking for different tools that help to develop more effective national and global climate policies, by fully accounting for international trade. ERAs may prove helpful since schemes for credits and penalties should be developed for emissions across all countries to fully account for trade in emissions. For example, the Climate Change Act of Scotland[34] not only aims to account for PBA emissions but also CBA. The reduction of global GHG emissions could well be based on and valued in terms of the effectiveness of national and international actions as measured by ERAs.

So far, we have focused on the ex post features of ERAs, trying to emphasize how ERAs may be used to assess previous and current agreements. However, ERAs may also lead to ex ante suggestions to guide trade policy, providing the right signals toward reducing emissions. Ultimately, they can indicate what countries should do in terms of choosing/prioritizing actions that lead to more global emissions reductions (e.g., with a certain partner or with a bilateral agreement). This seems particularly relevant in the context of the Paris Agreement[11] (Decision paragraph 20 and Article 14), which requires tracking the progress of individual and collective contributions every 5 years[35].

ERA adapts CBA. Most of the advantages and disadvantages of CBA hold therefore also for ERA. The advantages that are typically brought forward include: political benefits, more equity, and justice[36–38], and providing a basis for border carbon taxes or adjustments[39]. Disadvantages of CBA, and implementations of the corresponding taxes and adjustments include concerns about effectiveness and efficiency, impediments of practical implementation, or political incompatibility[33].

We agree with some points of critique on CBA, e.g., the current limits to obtain fully tractable information—such as emission

multipliers—for specific products. However, we also think that other points of critique can be solved or reduced, and that CBA-based methods have potential to improve climate policy. For example, CBA was has been criticized by arguing that it does not necessarily provide a direct link between a country's actions and changes in emissions[40]. This is precisely an issue that is addressed and remedied with ERA.

Also, some points of critique do not only apply to CBA, e.g., concerns about the link between border carbon taxes or adjustments and the effects on emissions are concerns that apply to any tax. These concerns are very valid but relate to unavoidable limits, such as our incapability of predicting the future with certainty. ERA provides the key elements to analyze scenarios ex ante and to evaluate the effects ex post. Further points of critique are related to compiling the data and taking modeling decisions (e.g., on the regional and sectoral disaggregation). These points are related to the reliability and transparency of information. However, as we argue next, important improvements have occurred in this respect.

The implementation of CBA and ERA requires GMRIO data linked to emission data at the same detail level. For a long time, global and comparable (notably across time) data, at a policy-relevant level of aggregation of industries and products, were lacking. At the same time, there are the usual limitations of IO models, such as the assumption that one industry produces exactly one product implying that secondary products just do not exist[41–44]. Also specific limitations of IO analysis exist for emission accounting. Examples are: the consequences of biases due to the aggregation of products and countries; the misalignment of IO tables with energy-emission methodologies and accounts; or the treatment of the transport sector and the valuation therein.

In recent years, however, more datasets have become available. Although IO analysis originates from the field of economics, it is noteworthy that the first global IO databases were developed in the environmental sciences. Such databases are now gathered and updated in a consistent and timely manner (WIOD[14], EXIOBASE[45–47], EORA[48], OECD[49], and GTAP[50,51]), given their own underlying philosophies[52]. These databases allow for more cross-country and temporal comparisons of the changes in GHG emissions[17,22,53]. Furthermore, it has been showed that the main results converged across GMRIOs for carbon footprint accounting and disparities were basically due to different definitions of the environmental stressor used[19] (also the differences between GMRIOs have been analyzed[54]). We expect more studies that: compare (or simulate, or estimate) projected energy and emission mitigation pathways over time;[7] assess—as required in the Paris Agreement—the compliance with (or deviations from) the national targets (NDCs); and value the ITMOs, and national and global objectives and projections. ERAs and the underlying scheme of credits and penalties can be very useful for these studies.

ERAs or a similar system of credits and penalties may not be implemented in the very short run, but it certainly seems a viable option for the not so distant future. In the same fashion that a supranational institution like the OECD started to produce GMRIO tables, there are interesting movements into the direction of generalizing accounting systems. For example, OECD and WTO joining forces in their initiatives to measure trade in value added or TiVA[55]. Another example is the Project Réunion, in which a group of researchers (representing the existing GMRIO datasets) met to discuss the way forward, which was an important incentive for forming the virtual MRIO labs[52,56].

Convergence of GMRIO tables[19,54] suggests two options for researchers and policy makers. On the one hand, if the results with separate datasets do not differ very much, one may safely work with average results. On the other hand, a supranational

institution (e.g., the UNFCCC) may decide to pick a single database, and require that it is used for tracking the developments and evaluate the policies. This tracking and evaluating is considered a must by many authors and was incorporated in the Paris Agreement. It is also where ERAs may prove to be useful.

In the discussion on (shared) producer/consumer responsibilities, many authors emphasized that both perspectives need to be considered[38]. We have chosen to have ERA adapt CBA. First, because CBA adapts PBA by taking trade and global emissions into consideration. Second, because ERA includes the next step by taking account of different trading possibilities and incentivizing to trade better or smarter. Supplementary Note 4, however, shows how a scheme of credits and penalties can be developed in a framework of shared responsibilities. Further developments could consider modified forms of the consumer perspective approach, e.g., incorporating capital stock changes[57], or based on MNE foreign affiliates' responsibility[58].

Several authors view implementation and strengthening of the Paris Agreement as next steps in the global response to climate change[59]. Any international negotiation about assigning burdens (or distributing efforts, or sharing responsibilities) will be dominated by basic considerations[60,61]. These include criteria for equity and fairness, historical responsibilities, and the countries' capacity to pay. The role for ERA would—in our view—be to complement these basic considerations and do what it does best, quantifying the environmental effects of recent past or projected future developments.

## Methods

**Standard emission accounting methods**. This section presents the standard PBA and CBA, TCBA (proposed by Kander et al.[4]) and TCBA* (TCBA with the modification proposed by Domingos et al.[5], implemented in Kander et al.[62]). Supplementary Note 1 of the SI gives the full methodological framework for PBA, CBA, TCBA, TCBA*, and ERA. Here, we only present the main equations.

If one aims at reducing global emissions, one would like to develop a scheme to credit actions by one or more countries that reduce global emissions and penalize actions that increase global emissions. It is well known that CBA does not satisfy in this respect and sends out perverse stimuli. This led Kander et al.[4] to adjust the original CBA and remedy this shortcoming. The example in Supplementary Note 2 of the SI shows, however, that also TCBA and TCBA* may penalize a country that engages in trade that reduces global emissions.

The emission intensities $g_i^R$ give the emissions per unit (e.g., dollar) of production in industry $i$ ($=1, \ldots, n$) of country $R$ ($=1, \ldots, N$). The element $l_{ij}^{RS}$ of the $Nn \times Nn$ matrix $\mathbf{L}$ gives the production in industry $i$ of country $R$ that is necessary for one dollar of consumer demand for final product $j$ produced in country $S$. The emission multiplier is given by $\sum_R \sum_i g_i^R l_{ij}^{RS}$ and tells how much is emitted globally for one dollar of consumer demand for final product $j$ produced in country $S$.

The PBA (minus the emissions directly by households) for country $R$ is given by

$$\text{PBA}^R = \sum_i g_i^R x_i^R = \sum_i g_i^R \left( \sum_T x_i^{RT} \right) \tag{1}$$

where $x_i^R$ gives the production in industry $i$ of country $R$ and $x_i^{RT}$ gives the production in industry $i$ of country $R$ that is embodied in all consumer demand in country $T$ for final products. The CBA (minus the emissions directly by households) is for country $R$ given by

$$\text{CBA}^R = \sum_T \sum_i g_i^T x_i^{TR} = \text{PBA}^R - \sum_{T \neq R} \sum_i g_i^R x_i^{RT} + \sum_{T \neq R} \sum_i g_i^T x_i^{TR} \tag{2}$$

CBA equals PBA minus the export of domestic emissions plus the import of foreign emissions. For the TCBA, the domestic emission coefficients ($g_i^R$) are replaced in the exports of emissions by world market average emission coefficients ($\bar{g}_i$), with

$$\bar{g}_i = \frac{\left( \sum_S \sum_{T \neq S} g_i^S x_i^{ST} \right)}{\left( \sum_S \sum_{T \neq S} x_i^{ST} \right)} \tag{3}$$

This yields

$$\text{TCBA}^R = PBA^R - \sum_{T \neq R} \sum_i \bar{g}_i x_i^{RT} + \sum_{T \neq R} \sum_i g_i^T x_i^{TR} \tag{4}$$

Domingos et al.[5] propose to also apply the world market average coefficients in Eq.

(3) to the imports. In that case, the adapted TCBA becomes

$$(TCBA^*)^R = PBA^R - \sum_{T \neq R} \sum_i \bar{g}_i x_i^{RT} + \sum_{T \neq R} \sum_i \bar{g}_i x_i^{TR} \qquad (5)$$

**Emission responsibility allotments.** The example in Supplementary Note 2 of the SI shows that CBA and its adjustments TCBA and TCBA* suffer challenges as basis for a scheme of credits and penalties. The example considers a situation of pure Ricardian trade (with comparative advantage defined in terms of emitting the least $CO_2$) and finds that one of the trading partners is penalized while trade reduces global emissions. Therefore, we need an adapted framework and for this we propose to use the ERAs. If the aim is to reduce global emissions then any action (such as additional trade) that decreases (increases) emissions should be credited (penalized). Moreover, the larger the reduction in emissions the larger the credits. ERAs adapt CBA on the basis of the gains and losses for global emissions. The situation where the traded goods had been produced at home is used as a benchmark. Next, we present the method to determine ERAs after which we discuss desirable properties imposed on schemes of credits and penalties in section 3.

For ERAs, we start with the global emissions that are embodied in the final goods that are produced in R and consumed (or used as investments) in country S. This is to be compared with the case where the exports of emissions had not taken place. Instead, all final goods are assumed to have been produced at home in country S.

The situations to be compared are: first, the actual situation where country S buys its final products (for consumption and investment purposes) in country R; and, second, the hypothetical case in which country S had produced these final products at home. It should be stressed that the counterfactual only affects the trade in final products. In principle one could extend the analysis to include also the trade in intermediates. That is, instead of buying intermediates in country R, produce them at home in country S.

The central idea in Ricardian theory is that a country should export the goods and services in which it is best in terms of production. This holds, even if it is always worse than its trading partner. If countries trade in this way they both will gain from trade. Best is defined as using the least amount of the scarce resource under consideration. Traditionally, that was labor and more trade leads to increased welfare in both countries. Alternatively, however, one could take environmental aspects into consideration. For example, define best as generating the least amount of emissions in the production of a certain good or service[63], or using the least amount of water[64]. In that case, increased trade will reduce the emissions (or water consumption) in each of the two trading countries.

Final consumers (and clients in general) are a powerful force in persuading producers to act in a responsible way, as witnessed by the growing literature on corporate social responsibility. This also applies to their responsibility for the environment. Despite the problem of having constraints in MRIO on compilation product-specific data that might require time to see great accuracy and detail, through labeling, consumers can be informed how much $CO_2$ or GHGs are embodied in a certain final product. If the consumers' environmental concern is sufficiently large they may decide not to buy the cheapest alternative but the alternative that is produced the cleanest. Something similar has happened before with campaigns aiming to ban brands that sell final products embodying child labor[65,66]. According to Ballet et al.[67]: "the basic strategy for the fight against child labor has been boycotting efforts followed by labeling practices".

Bearing the role of consumers in mind, we have chosen to develop ERAs for the case of trade in final products. Yet, Supplementary Note 4, however, shows what the framework looks like if we develop a scheme of credits and penalties based on trade in intermediate products. This would directly incentivize producers to trade better.

Our scheme of assigning credits and penalties is based on the idea that it takes two to tango. Both countries are credited (penalized) equally if their bilateral trade decreases (increases) global emissions. The amount of the credit or penalty is determined by comparing the actual situation with the hypothetical situation that imports of emissions had been replaced by emissions at home. Consider two countries: R and S. Let $e_i^R$ indicate the emission multiplier, i.e., the global emissions that are generated somewhere in the production chain of one unit (say dollar) of final demand for good i from country R. Let $y_i^{RS}$ indicate the final demand in country S for good i from country R. The global emissions involved in the imports by S of final goods from R are given by $\sum_i e_i^R y_i^{RS}$. This is to be compared with the hypothetical situation in which these imports (by S from R) had been produced at home (i.e., in S). In that case, the global emissions would have been $\sum_i e_i^S y_i^{RS}$. If the difference $\sum_i (e_i^R - e_i^S) y_i^{RS}$ is negative (positive), it gives the reduction (increase) in global emissions and reflects the gains (losses) from the exports from R to S. Vice versa, the imports by R from S (which are equal to the exports from S to R) changes global emissions by $\sum_i (e_i^S - e_i^R) y_i^{SR}$. The extra emissions due to bilateral trade are given by

$$\sum_i (e_i^R - e_i^S)(y_i^{RS} - y_i^{SR}) \qquad (6)$$

and a negative outcome indicates reduction of global emissions.

Note that the gains and losses above were derived from the viewpoint of country R. That is, exports from R (to S) and imports by R (from S). The same answer is

obtained if the calculation is done from the viewpoint of country S, implying symmetry. We thus assign half of the gains or losses, i.e., $(1/2) \sum_i (e_i^R - e_i^S)(y_i^{RS} - y_i^{SR})$, to each of the two countries. The outcome points at a credit if it is negative and a penalty if it is positive. In the case of N countries, the changes in global emissions due to trade by country R are given by

$$(1/2) \sum_S \sum_i (e_i^R - e_i^S)(y_i^{RS} - y_i^{SR}) \qquad (7.)$$

It should be stressed that the outcomes are particularly relevant when their development over time is considered. The numbers themselves involve a comparison of the actual situation with a hypothetical no-trade case. Comparing the numbers over time, however, allows to analyze the effect of changes in trade (next to changes in emission efficiency and production technology).

A final adaptation makes the scheme satisfy the condition of additivity. The extra emissions due to bilateral trade between R and S are given by $\sum_i (e_i^R - e_i^S)(y_i^{RS} - y_i^{SR})$ and $(1/2) \sum_i (e_i^R - e_i^S)(y_i^{RS} - y_i^{SR})$ is assigned to each of the two countries. The average extra global emissions due to bilateral trade are given by

$$a = \left[ (1/2) \sum_R \sum_S \sum_i (e_i^R - e_i^S)(y_i^{RS} - y_i^{SR}) \right] / N \qquad (8)$$

as there are N countries. For country R, instead of using $CBA^R = \sum_S \sum_i e_i^S y_i^{SR}$, we propose to use the ERA defined as

$$ERA^R = CBA^R + (1/2) \sum_S \sum_i (e_i^R - e_i^S)(y_i^{RS} - y_i^{SR}) - a \qquad (9)$$

where the penalty is given by $(1/2) \sum_S \sum_i (e_i^R - e_i^S)(y_i^{RS} - y_i^{SR}) - a$. Equation (9) shows that country R is credited if its trade leads to a reduction of the global emissions that is larger than the average reduction due to trade. In that case is the penalty negative and do we have $ERA^R < CBA^R$. Our scheme of credits and penalties (as follows directly from ERA) is based on the gains in global emissions due to the trade in final products by country R (relative to the average country's gains in global emissions). As we will discuss in section 3, the scheme of credits and penalties satisfies all three desirable properties mentioned in Kander et al.[4].

In the Supplementary Note 1 (subsection 1.4) of the SI, we provide an illustration of the functioning of the method for three countries and two goods. We also extend the method to the industry level, which is illustrated in subsection 1.5 of Supplementary Note 1 in the SI.

**Desirable properties for credits and penalties.** Kander et al.[4] formulate three important and intuitively compelling properties for a scheme of credits and penalties. These three properties are a subset of the six properties listed in Rodrigues et al.[6] and Domingos et al.[5], albeit using a different terminology. The three properties in are[4]: sensitivity, monotonicity, and additivity. The additional three properties are[6]: economic causality, symmetry, and scale invariance. In our discussion, we start with the three properties and show that ERAs satisfy these properties. After that we indicate why economic causality and symmetry are not relevant in the present context. We also discuss why the requirement of scale invariance, which was at the heart of the comment by Domingos et al.[5], is not met by CBA, TCBA (nor indeed TCBA*), and ERA and why it may not be so desirable.

The first property, i.e., sensitivity, means that the credits and penalties are "responsive to factors that nations can influence" (Kander et al.[4]). Typical examples of such factors are: changes in final demands (not only levels but also the composition of the bundle) that decrease global emissions; decreases in emission intensities; and changes in the production structure (e.g., a larger productivity in the sense of using less intermediate inputs per unit of output, or replacing the import of intermediate inputs from a dirty country by imports from a clean country). Supplementary Note 3 of the SI provides the technical details and shows that ERAs are sensitive in the sense that they are responsive to factors that a country can influence and that affect global emissions.

The second property, i.e., monotonicity, states that a country should not be able to reduce its own carbon responsibility by increasing global emissions[4]. So, a country should not be credited for actions that increase global emissions or penalized for actions that reduce global emissions. This is a very strict definition, using increases and decreases in absolute terms. ERAs and the underlying scheme of credits and penalties do not satisfy this property. However, they are constructed to satisfy a slightly weaker form of monotonicity that uses changes in global emissions in relative terms. Countries are credited (or penalized) if their national actions reduce global emissions more (or increase them less) than average. Also, the larger the reduction in global emissions the larger the credit (or the smaller the penalty).

The third property is additivity and requires that "the sum of national emissions for all countries should equal total global emissions" (Kander et al.[4]). Recall that Eq. (8) introduced the average extra global emissions due to bilateral trade. The ERAs in Eq. (9) corrected the CBA for a reduction in global emissions due to trade that is larger than the average reduction due to trade. The initial set of credits and penalties was rescaled to make the sum of all responsibilities equal to the global emissions. This led to additivity.

When discussing the three additional properties (economic causality, symmetry, and scale invariance), it should be emphasized that the approach in Rodrigues et al.[6] is

slightly different. For any economic flow, they distinguish between its upstream and its downstream environmental pressure. Gallego and Lenzen[3] provided a comprehensive framework on the upstream and downstream perspectives[68]. CBA (and TCBA and ERA) only considers an upstream perspective and simply has no downstream perspective. Economic causality requires a proportional link between the upstream and downstream environmental pressure and an economic flow. Symmetry requires that the upstream environmental pressure of flow ($i$, $j$) equals the downstream pressure of flow ($j$, $i$). Given the different viewpoint on (re)distributing environmental pressures and the role of distinguishing between upstream and downstream pressures, neither economic causality nor symmetry is relevant in the present context (with only an upstream perspective).

It should be mentioned that a different type of symmetry does apply in the case of ERAs. That is, the gains and losses (in terms of changes in global emissions) of bilateral trade between countries $R$ and $S$, are the same no matter whether they are evaluated from the perspective of $R$ or of $S$. Both perspectives yield the same answer. The credit or penalty that follows from the gain or loss is the same for both trading partners if they share the consequences equally.

In the comment by Domingos et al.[5] and the response by Kander et al.[62], the focus is on the last of the remaining properties, namely scale invariance. It means that, for any union of countries, the sum of emission responsibilities for all countries in the union must equal the emission responsibility of the union if it were treated as a single country. Domingos et al.[5] argue that TCBA does not satisfy the scale invariance condition and come up with an alternative.

We argue that CBA, TCBA, but also the method proposed by[5] and our own ERA do not satisfy scale invariance. The reason is that any method in IO analysis that is based on the Leontief inverse suffers from aggregation bias. The underlying idea is as follows. At the aggregate level, one may provide two answers to a single question. One answer is obtained from aggregation after calculation, the other answer from calculation after aggregation. Suppose we have a large GMRIO table and suppose that a number of individual countries form a union (like EU27 in the WIOD tables). In the case of aggregation after calculation, we first calculate the answer using the full GMRIO table and then aggregate the answers for the individual countries in the union to arrive at the answer for the union. In the case of calculation after aggregation, we first aggregate the GMRIO table to arrive at a smaller table that includes the union as if it were a single country. Next, we run the calculations on this aggregated table that yields answers for individual countries, one of which represents the union. These two types of calculation yield the same answer only under very strong properties that do not hold in real world cases, or by sheer coincidence[41]. In general, the answers will differ and the difference is termed aggregation bias. Because the Leontief inverse **L** is a non-linear function of the input matrix **A** ($\mathbf{L} = (\mathbf{I} - \mathbf{A})^{-1} = \mathbf{I} + \mathbf{A} + \mathbf{A}^2 + \dots$), questions for which the answer involves the use of **L** suffer from aggregation bias. This raises the question whether scale invariance is such a desirable property. If it is, the accounting framework cannot be based on the IO model, because this model uses **L** that makes the framework scale variant. PBA is an example of an accounting framework that satisfies the condition of scale invariance (and that—indeed—does not use an IO model).

**Reporting summary**. Further information on research design is available in the Nature Research Reporting Summary linked to this article.

## Data availability

The authors declare that all the data (as well as the code) supporting the findings of this study are available within the paper and its supplementary information files, as follows: the source data underlying all the figures and tables in the manuscript are provided as Supplementary Data files: [https://data.mendeley.com/datasets/2hvsgsfw3z/2]. The WIOD data (release 2013) that support the findings of this study is available in the website [http://www.wiod.org/release13].

## Code availability

The codes used in this text to perform the PBA, CBA, TCBA, TCBA∗, and ERA analysis are provided as Supplementary Data files: [https://data.mendeley.com/datasets/2hvsgsfw3z/2].

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

## Acknowledgements

I.A. and I.C. thank the support of the Spanish Ministry of Science, Innovation, and Universities, through the project Modeling and analysis of low carbon transitions (MALCON, RTI2018-099858-A-I00), and the Spanish State Research Agency through María de Maeztu Excellence Unit accreditation 2018-2022 (Ref. MDM-2017-0714) and Basque Government BERC Programme.

## Author contributions

E.D. designed the methodology and I.A. and I.C. contributed to background and analysis. All authors contributed to writing the paper.

## Competing interests

The authors declare no competing interests.
