## [Peer Review File · Nature Communications]

Reviewers' comments:

Reviewer #1 (Remarks to the Author):

Please find here the comments to the authors about the proposed manuscript NCOMMS-17-26938, entitled "Towards a more effective climate policy on international trade". The ambition of the paper is to propose a new scheme for assign emissions responsibilities among countries, considering a credits and penalties program according to technology differences. In my opinion, this a necessary contribution in the literature because is looking for alternatives moreover the predominant Producer Responsibility criterion, the only commonly accepted by the international community.

Although I do not agree with some of the assumptions, I think this new approach could result useful and could reinforce the debate about the necessity to take into consideration the technology differences among countries when emissions are internationally traded and the responsibility is allocated. The methodology proposed, the main contribution of the paper, is perfectly explained, it is really easy to read and understand, and fits perfectly with the literature of reference. That is why I think the paper is suitable for publication in Nature Communications.

However, I have few concerns about some of the main assumptions, about the structure of the paper, the absence of results analysis and with the discussion section that I would like to discuss with authors. In the following lines, I will try to identify minor, and in my opinion, major concerns about the paper.

Misprints or minor changes:

- Line 53: remove the square brackets in the word "actions".
- Line 336: change BCA by CBA.
- Line 350: remove the square brackets in the word "countries".
- Line 356: remove the ellipsis after global.
- Line 375: the reply to Domingos et al. suggested in this paragraph is not the reference number 7; moreover, it is not listed in the references section. Reference: Kander, A., M. Jibron, D. D. Moran, and T. O. Wiedmann. 2016. Reply to 'Consistency of technology-adjusted consumption-based accounting'. Nature Clim. Change 6(8).

General comments on the paper:

- Classical Ricardian trade theory: since the abstract, you link constantly your contribution to the Ricardian theory, and I am not sure about the robustness of the link. You perfectly define the idea of the theory (lines 181 to 186), but you link it with a future hypothetical case based on the consideration of "the best" as the generation of the least amount of emissions. In this line, regarding trade and emissions transfers please read and revise Jakob and Marchinski 2012 (Jakob, M. and R. Marschinski. 2012. Interpreting trade-related CO2 emission transfers. Nature Climate Change Online,

23 September). Currently, one could consider the opposite case; high levels of environmental pressures have so far been understood, in some cases, like emerging economies comparative advantages, because of the related competitive energy costs. In the current situation, with an under reform EU-ETS scheme and with the Producer Responsibility criterion as the only one on the table of the COP23, the approach could be too far from reality.

- Literature review: I recommend reading the paper from Raupach et al. 2014, in Nature Climate Change, because there are more alternatives to share cumulative emissions than among countries. I think this paper could be useful to improve the quality of the conclusions/discussion of this paper.

Regarding the estimation of balances of avoided emissions, the concept behind your ERA proposal, has been widely discussed in the recent literature. Please read López et al. (2014) and Zhang et al. (2017), the conclusions could be useful in your sections 4, results, and 6, discussion. References: López, L. A., G. Arce, and J. E. Zafrilla. 2013. Parcelling virtual carbon in the pollution haven hypothesis. *Energy Economics* 39(0): 177-186.; Raupach, M. R., S. J. Davis, G. P. Peters, R. M. Andrew, J. G. Canadell, P. Ciais, P. Friedlingstein, F. Jotzo, D. P. van Vuuren, and C. Le Quéré. 2014. Sharing a quota on cumulative carbon emissions. *Nature Climate Change* 4: 873; Zhang, Z., K. Zhu, and G. J. D. Hewings. 2017. A multi-regional input-output analysis of the pollution haven hypothesis from the perspective of global production fragmentation. *Energy Economics* 64: 13-23.

- 3.2 example: the example of the 3.2 section is useful to replicate, systematically, the method proposed using a simple MRIO model, however, in my opinion, it should be placed in the Supporting information section. This is only an invented example (I do not know if it has been arbitrarily created by authors), so the contribution, which is only to know how the approach works, is a complementary information, it should not be part of the main manuscript. Besides, the explanation of the methodology in section 3.1 is almost perfect, so I think the example is not required there.

- Regarding ERA proposal: In terms of equity, and considering the similar result of ERA compared to CBA, the most penalties would fall to the imports from rich countries that in many cases are generated in emerging less efficient countries. This fact could imply a disincentive for the developing of emerging economies because of the rise of products prices. A possible solution would be the one proposed by Springman 2014 (Springmann, M. 2014. Integrating emissions transfers into policy-making. *Nature Clim. Change* 4(3): 177-181): the potential environmental border taxes revenue could be used to incentive the transition to more efficient emerging countries. In fact, this could be considered as an Internationally Transferred Mitigation Outcome (ITMO) suggested in the Paris Agreement.

If we consider ERA as the best criterion, you should address how to implement it. How countries are going to transfer the responsibility to industries and consumers. The full allocation of responsibilities to the consumer industries would imply financial problems in some industries, like the Textile, as is shown in Cadarso et al. 2012 (Cadarso, M.-Á., L.-A. López, N. Gómez, and M.-Á. Tobarra. 2012. International trade and shared environmental responsibility by sector. An application to the Spanish economy. *Ecological Economics* 83: 221-235). Moreover, the estimation of the penalty is based on the total amount of emissions related to international trade. Why do you not estimate ERA comparing differences (domestic - imported) by sectors averages? How would the responsibilities change?

Also in terms of equity, it could be also useful to analyze the increases or reductions of responsibilities after the application of the ERA criterion, compared to PBA and CBA, by a list of countries organized by per capita income. This would be helpful to know whom, clean rich or dirty emerging countries, are more benefited. Is there any relationship with the level of development?

- MRIO vs BRIO: the use of the MRIO model to estimate ERA in front of CBA, makes countries responsible for their consumption (but not for production). Subsequent emissions reductions commitments have to be transferred to industries. In this line, you have to punish or credit industries, so it seems also interesting to compare production technologies more than consumption patterns. All those considerations bring me to consider the use of a Biregional input-output model (BRIO) instead of an MRIO model. Using this approach, you compare the domestic efficiency of the importer country, with the domestic efficiency of the exporter. This would be helpful also to provide better evidence about the industries at risk of carbon leakage to reformulate the forthcoming revisions of the EU-Emissions Trading Scheme. Have you considered the application of a BRIO model? Have you thought about the pros and cons of it?

- Results: There are no results in section 4. In my opinion, although your paper could be considered as a theoretical approach, the empirical dimensions of the consequences of the ERA criterion have to be analyzed. I recommend you to provide a deep analysis of the ERA by countries, years, comparison with the rest of approaches, etc. And, of course, this has to be presented using more attractive figures, tables and diagrams, as I will recommend you in the next comment.

- Results visualization: the results visualization is very poor. It is in line with the scarcity of the results analysis. I consider that better and more readable figures and tables are needed. Please, use better tools, colors, change the thickness of lines, rescale axis, etc. to improve the readability and to identify different patterns, behaviors, trends, etc. Consider the possibility to change the Table 1 (or place it in the Supporting Information) and use another type of graphic to illustrate the real differences between criteria and countries (who is more credited? who is the most penalized?). In this sense, the use of a kind of Sankey diagram or CIRCOS graphic presenting figures related to the size of penalties and the origin of the most penalized trade relationship could be useful in order to know who are the winners and losers of ERA (I follow the recommendations suggested in previous comments).

- Desirable properties, section 5: the discussion of the properties is always interesting, as is possible to read in Kander et al. (2015), Domingos et al. (2016) and reply. That is why I consider necessary its inclusion in this paper. However, section 5, after results, is not the best place to insert it. In my opinion, it should be placed in section 3 where the ERA is proposed.

- Discussion and policy implications: I consider that this discussion must be focused in the adequation of the new proposal in the international context. Moreover, subsequent policy implications derived from the application of ERA criterion would be explained in detail. In my opinion, the description of the Kyoto Protocol mechanisms is not useful in this context, Kyoto is the past, and the justification of the alignment of ERA with the Paris Agreement tools (ITMOs) is not clear. How ERA helps the instrumentalization of ITMOs like SO₂ trading systems in the USA, the new EU-ETS (considering that all the countries should have their own cap-and-trade programs) or promotes Clean Development Mechanisms? The way you would implement it (via Carbon Taxation using mechanisms like those proposed in Springmann 2014?) should be discussed here, in

comparison with the absence of alternative incentives following the PBA or CBA criteria. The results show by ERA are really close to CBA, so is hard to imagine why here's proposal is in line with the recommendations of Article 6 of the Paris Agreement, considering that PBA was proposed as the allocation criterion. In fact, you propose this idea since the abstract, and I cannot identify a clear relationship. Maybe you could try to clarify the relationship.

Reviewer #2 (Remarks to the Author):

The article is in general well written, the points well argued, and the results and message relevant for the climate change community. The article is perhaps somewhat too much on the technical side for Nature Communications, but perhaps this is unavoidable given the nature of the issue at hand.

One main concern is that the significance of the work doesn't really jump at you. In the introduction, the reader must get a feeling not only for why this may be new and different to what has been done before, but also for why this is important? What is actually at stake? Differences of hundreds of megatonnes depending on how you account? Negotiation strategies depending on this? Etc etc.

o The authors mention (in the intro) Gallego's shared responsibility approach (using eg value added shares as responsibility weights), but then IO cannot see this taken up anywhere in the analysis (eg at the outset of section 2). Why not? Already the sentence that follows the mention continues dealing with PBA and CBA only.

o "the properties of "sensitivity", "monotonicity" and "additivity" " need to be explained for the lesser prepared reader – this is not straightforward. I know this comes in section 5, but it should be motivated at least to not leave the reader hanging.

o I understand that the mathematical content of section 2 is needed. However, for the more general readership of Nature Communications, I would farm this out to an Appendix and present the same content in a more accessible form. This may be challenging, but would benefit the article if done well. The same applies to Section 3.1.

o The authors say that the symmetry criterion is not relevant in an assessment confined to an upstream perspective (if I understood this right), but they could perhaps mention that an extension to a downstream perspective is straightforward, and already laid out by Gallego and also in Lenzen, M. & Murray, J. Conceptualising environmental responsibility. Ecological Economics 70, 261-270 (2010).

Minor comments:

- typo: "should credit actions that contribute to reduced global emissions AND SHOULD penalize actions that increase them"

- This is because foreign emissions are embodied in the intermediate inputs – in an MRIO framework, aren't there also direct imports into final demand that carried emissions embodiments, since y is a matrix?

- "We argue that emission accounting is one thing (as PBA and CBA do)"- the grammar in that sentence is not right.

Reviewer #3 (Remarks to the Author):

The manuscript by Erik Dietzenbacher et al. „Towards a more effective climate policy on international trade“ proposes a scheme for assigning responsibilities for emissions among countries accounting for trade interdependencies. In particular, the scheme credits actions that reduce global emissions and penalize actions that increase them. The manuscript take up a discussion that has been taking place in Nature Climate Change (Kander et al., 2015; Domingos et al., 2016) and, in fact, provides important clarifications and corrections. The manuscript is well written and I could not find problems in the underlying math. I still reject this manuscript, because I do not perceive it to be of interest to a broad readership as served by Nature Communications. It should be submitted to a more specialized journal (which, in fact, would also better suit the current structure of the manuscript).

Having said that, I would like to qualify that I would equally have rejected the pieces by Kander et al. (2015) and Domingos et al. (2016) on the same grounds (despite their additional problems). I provide this qualification for transparency to enable the editors to over-rule my judgement in case of disagreement.

1) Relevance of consumption-based accounting in the context of the broader discussions on responsibility in international climate policy:

There is a wider discussion on assigning responsibility for greenhouse gas emissions and anthropogenic climate change that goes far beyond consumer and producer responsibility. Accounting for trade in the assignment of responsibility is usually motivated in terms of developing countries as their producer emissions are much higher than their consumer emissions. Yet, the difference between consumer and producer responsibility is dwarfed by extending the relevant time frame back to pre-industrials as proposed in the discussion around historical responsibility. Of course, one discussion does not exclude the other – particularly, given the fact that, in principle, historical trade-adjusted emissions accounts could be developed (even though this is a theoretical argument as constructing reliable accounts seem impossible). Yet, trade-adjusted accounts have remained a small academic discussion and not received much attention in international climate policy, while historical responsibility has been a key concern in the negotiations for many developing

nations from the beginning. In my view this will not change in the future – partly due to the point I will explain next.

2) Input-Output Analysis as basis for certified and verified emissions reductions: For me one fundamental question in this entire discussion is whether we believe that political consensus could be achieved on such a scheme and its methodological and data foundation. Put in a different way, do we expect that the proposed scheme could be the basis for certifiable and verifiable emission credits or penalties? I think this is far from perceivable. Trade-adjusted accounts required to make the necessary calculations rely on input-output analysis. The various error sources have been discussed extensively in the scientific literature (e.g. Lenzen et al., XXX). The authors themselves, for example, point towards the fact that aggregated nature of input-output tables make it impossible for the method to be scale-invariant. In general, aggregation bias highlights that the level of aggregation of the input-output calculus partially determine the results. This makes the tool impractical as a basis for verified emission credits or penalties. While input-output tables work usually at the level of dozens or at best a few hundreds of sectors there are ten-thousands of products that are internationally traded. In the absence of reliable product level accounts in the foreseeable future, I perceive this as a merely academic discussion that can influence climate policy at best on a rhetorical level.

3) Structure of the manuscript: The manuscript with its focus on methodological details is written for a field journal rather than a broader readership. Of course, this could be changed. If accepted, this needs to be changed. Most of the details of sections 2 and 3 should be shifted to the Supplementary Information. Results could be discussed in more depth. The discussion would need to be extended to discuss relevant limitation of the proposed method and link this to requirements for application in political processes like the UNFCCC.

Some specific comments on the manuscript: Just a few additional comments.

- As highlighted before a discussion of the limitations of input-output analysis for emission accounting is missing (various biases).
- Figures all lack axis labels
- Data description in my view is insufficient. We have, for example, no idea what CO₂ emission component this is – as it is obviously not all anthropogenic CO₂. How does it compare to standard accounts (IEA, BP, EDGAR etc.)?
- No equation numbers make things cumbersome.
- Equation on page 7: It is entirely unclear why changes in emissions should be equally shared by countries. See Gallego and Lenzen (2005).

Literature:

Domingos T, Zafrilla JE, Lopez LA. Consistency of technology-adjusted consumption-based accounting. *Nature Clim Change* 6, 729-730 (2016).

Gallego, B., Lenzen, M., 2005. A consistent input–output formulation of shared consumer and producer responsibility. *Economic Systems Research* 17 (4), 365–391.

Kander A, Jiborn M, Moran DD, Wiedmann TO. National greenhouse-gas accounting for effective climate policy on international trade. *Nature Clim Change* 5, 431-435 (2015).

Lenzen, M. 2000. Errors in conventional and input-output-based lifecycle inventories. *Journal of Industrial Ecology* 4(4): 127–148.

Reviewers'

comments:

Reviewer #1 (Remarks to the Author):

We thank the referee very much for his/her very useful comments. Below, we address point-by-point the suggestions and indicate the changes they have induced.

Comment: Please find here the comments to the authors about the proposed manuscript NCOMMS-17-26938, entitled "Towards a more effective climate policy on international trade". The ambition of the paper is to propose a new scheme for assign emissions responsibilities among countries, considering a credits and penalties program according to technology differences. In my opinion, this a necessary contribution in the literature because is looking for alternatives moreover the predominant Producer Responsibility criterion, the only commonly accepted by the international community.

Although I do not agree with some of the assumptions, I think this new approach could result useful and could reinforce the debate about the necessity to take into consideration the technology differences among countries when emissions are internationally traded and the responsibility is allocated. The methodology proposed, the main contribution of the paper, is perfectly explained, it is really easy to read and understand, and fits perfectly with the literature of reference. That is why I think the paper is suitable for publication in Nature Communications.

However, I have few concerns about some of the main assumptions, about the structure of the paper, the absence of results analysis and with the discussion section that I would like to discuss with authors. In the following lines, I will try to identify minor, and in my opinion, major concerns about the paper.

Misprints or minor changes:

- Line 53: remove the square brackets in the word "actions".
- Line 336: change BCA by CBA.
- Line 350: remove the square brackets in the word "countries".
- Line 356: remove the ellipsis after global.
- Line 375: the reply to Domingos et al. suggested in this paragraph is not the reference number 7; moreover, it is not listed in the references section.

Reference: Kander, A., M. Jibron, D. D. Moran, and T. O. Wiedmann. 2016. Reply to 'Consistency of technology-adjusted consumption-based accounting'. Nature Clim. Change 6(8).

Answer: We have adopted all changes and adapted the text accordingly.

Comment: General comments on the paper:

- Classical Ricardian trade theory: since the abstract, you link constantly your contribution to the Ricardian theory, and I am not sure about the robustness of the link. You perfectly define the idea of the theory (lines 181 to 186), but you link it with a future hypothetical case based on the consideration of "the best" as the generation of the least amount of emissions. In this line, regarding trade and emissions transfers please read and revise Jakob and Marchinski 2012 (Jakob, M. and R. Marschinski. 2012. Interpreting trade-related CO2 emission transfers. Nature Climate Change Online, 23 September). Currently, one could consider

the opposite case; high levels of environmental pressures have so far been understood, in some cases, like emerging economies comparative advantages, because of the related competitive energy costs. In the current situation, with an under reform EU-ETS scheme and with the Producer Responsibility criterion as the only on the table of the COP23, the approach could be too far from reality.

Answer: Thank you for the comments and for pointing out literature on trade and emissions transfers. We understand that low energy costs *have* been a source of better positioning of economies (being more competitive in at least one aspect). This is perfectly in line with the Ricardian theory (or the Heckscher-Ohlin-Vanek theory and models) if the aim is to minimize energy costs. This implies that future discussions need to be directed more towards the key question: what is it that we want to optimize? This could be: minimize the cost of production, or of energy use, or of labour use, but also minimize global emissions. We realize that in the not too distant future, traditional cost aspects (production, energy, labour) will prevail. However, two remarks in this respect.

First, we would like to point out (and quantitatively underpin) that it is possible to build trade on minimizing global emissions. Although our considerations are at this very moment still philosophical or conceptual in nature, we feel it is important that people realize that much of the discussion on GHG emissions and climate change is based on choices. And we feel it is important to show what might be the effects of making different choices.

Second, including different objectives in our thinking about climate change is not entirely unrealistic. The literature on marketing and social psychology shows a serious increase in research directed towards changing consumer behaviour. Also, consumers have become more powerful due to the boom in communication channels and tools. The campaign against the use of child labour shows that it is possible to change consumer behaviour.

We have tried to make all this more explicit and clearer.

The reviewer also rightly points out that “under reform EU-ETS scheme and with the Producer Responsibility criterion as the only on the table of the COP23”, methods which are based on some type of consumption criteria are unlikely to be implemented in the short term. We agree with the reviewer, but would like to point out two aspects (that are also related to the previous two remarks):

- We (and many others) feel that in climate policy it is necessary that actions (and their development and impacts) are accounted and tracked. For this, it is important that a method is used that is consistent, robust, without double counting. Also, the outcomes of applying this method should reflect reductions in emissions because of changes in production, consumption and trade. Measurement, reporting and verification (MRV) of greenhouse gas mitigation is key in climate action, not only for the emission inventories, but also for mitigation actions and their support. Particularly, the Paris Agreement requires tracking and documenting the progress of individual and collective contributions. Global stocktaking will take place every five years (UNFCCC, 2015, Decision paragraph 20 and Article 14), but the exact modalities of the review remain to be determined in the years to come (Röser, Fekete, Höhne, & Kuramochi, 2015).

- Even if the focus on ‘clean imports’ is not likely to be established in the short run, we think that we should not consider it impossible or unlikely in the medium term. For example, a few years ago, other compromises or the involvement of certain countries in agreements were also seen as unlikely. As we discussed in other replies and now more in the text, the method is not aimed to serve as a full system of accounting responsibilities entering into the debate of cumulative emissions, historical responsibility, etc. However, we claim that ERA –if it is adopted– can serve to guide or assign incentives and to evaluate the recent past based on the evolution of emissions. Also, recall that the Paris Agreement contains a range of principles which apply when Parties intend to use cooperation mechanisms to achieve their NDCs (e.g. the ITMOs, as we discuss in the paper). We think they cannot be properly tracked or evaluated without a method like the one proposed.

Comment: - Literature review: I recommend reading the paper from Raupach et al. 2014, in Nature Climate Change, because there are more alternatives to share cumulative emissions than among countries. I think this paper could be useful to improve the quality of the conclusions/discussion of this paper. Regarding the estimation of balances of avoided emissions, the concept behind your ERA proposal, has been widely discussed in the recent literature. Please read López et al. (2014) and Zhang et al. (2017), the conclusions could be useful in your sections 4, results, and 6, discussion.

References: López, L. A., G. Arce, and J. E. Zafrilla. 2013. Parcelling virtual carbon in the pollution haven hypothesis. *Energy Economics* 39(0): 177-186.; Raupach, M. R., S. J. Davis, G. P. Peters, R. M. Andrew, J. G. Canadell, P. Ciais, P. Friedlingstein, F. Jotzo, D. P. van Vuuren, and C. Le Quéré. 2014. Sharing a quota on cumulative carbon emissions. *Nature Climate Change* 4: 873; Zhang, Z., K. Zhu, and G. J. D. Hewings. 2017. A multi-regional input-output analysis of the pollution haven hypothesis from the perspective of global production fragmentation. *Energy Economics* 64: 13-23.

Answer: Thank you for the reference to the paper of Raupach et al. 2014 on alternatives to share cumulative emissions, and of those on the estimation of balances of avoided emissions. We have read and made use of some of them to enrich the conclusions/discussion of this paper. In particular, from (Raupach et al., 2014) we confirm the idea that different sharing responsibility/burden of reductions principles exist, and the complexity in factors affecting the decision (delay in mitigation, persistence, etc.).

Regarding the literature on the balances of avoided emissions, we have cited them. Still we consider that conceptual and methodological differences exist across approaches (e.g. as pointed out by the referee, the use of multi-regional vs. bi-regional computation) and goals. Also for the analysis of the pollution haven hypothesis, we consider that one could look more into the *emissions in exports*, while for this case, and in general for emission accounting, one should be more interested in the *exports of emissions* (emissions of one country that are driven by the final demand of others).

Comment: - 3.2 example: the example of the 3.2 section is useful to replicate, systematically, the method proposed using a simple MRIO model, however, in my opinion, it should be placed in the Supporting information section. This is only an invented example (I do not know if it has been arbitrarily created by authors), so the contribution, which is only to know how the approach works, is a complementary information, it should not be part of the main manuscript. Besides, the explanation of the methodology in section 3.1 is almost perfect, so I think the example is not required there.

Answer: we have moved the (yes, invented by the authors) example (previously in Section 3.2) to the Online Supplementary Information.

Comment: - Regarding ERA proposal: In terms of equity, and considering the similar result of ERA compared to CBA, the most penalties would fall to the imports from rich countries that in many cases are generated in emerging less efficient countries. This fact could imply a disincentive for the developing of emerging economies because of the rise of products prices. A possible solution would be the one proposed by Springman 2014 (Springmann, M. 2014. Integrating emissions transfers into policy-making. *Nature Clim. Change* 4(3): 177-181): the potential environmental border taxes revenue could be used to incentive the transition to more efficient emerging countries. In fact, this could be considered as an Internationally Transferred Mitigation Outcome (ITMO) suggested in the Paris Agreement.

If we consider ERA as the best criterion, you should address how to implement it. How countries are going to transfer the responsibility to industries and consumers. The full allocation of responsibilities to the consumer industries would imply financial problems in some industries, like the Textile, as is shown in Cadarso et al. 2012 (Cadarso, M.-Á., L.-A. López, N. Gómez, and M.-Á. Tobarra. 2012. International trade and shared environmental responsibility by sector. An application to the Spanish economy. *Ecological Economics* 83: 221-235). Moreover, the estimation of the penalty is based on the total amount of emissions related to international trade. Why do you not estimate ERA comparing differences (domestic - imported) by sectors averages? How would the responsibilities change?

Answer: We comment on the different points separately.

Regarding "it could imply a disincentive for the developing of emerging economies because of the rise of products prices", we agree, this is what could happen. The creation of a disincentive (to move domestic production to other countries that emit more) is what a credit/penalty system would signal. As society, we consider that emerging economies need support for their development, to accomplish the Sustainable Development Goals. For this, they certainly will need -and should receive- many different kinds of support. In our view, climate policy should not try to include other goals simultaneously. Rather, focus on the main priority (climate mitigation) and deal with possible negative consequences in a subsequent step.

The reviewer makes a very good point regarding "How to implement ERA, and transfer the responsibility to industries and consumers?" (our own representation of the reviewer's comments). Clearly, this problem exists already

now, and solutions tend to be varied in terms of actions, laws, constraints, etc. within countries. Even if –as pointed out by Steininger et al., 2014– one is concerned with the tax (or other measure) incidence rather than the tax (or other measure) base, any implementation tends to be disputed by the agent who is taxed. Policies such as favourable legislation for Energy Intensive and Trade-Exposed sectors have been criticized. In particular, with respect to the fairness of these regulations and their effectiveness in providing less stringent regulations for sectors that –although they may trade relevantly or being a crucial sector for the country– actually contaminate a lot. Although the specific design and implementation may take several directions, we view "penalties" imposed on both the production (industries) and consumption side (households) as the most logical and consistent with what ERA computes. The results in the paper focus on figures at the country level. It should be stressed that the computation method allows for more granularity of results (see the new Section A5 of the Online Supplementary Information). In this way, the responsibility can be transferred to industries. With respect to consumers, it is also logical to think that implementations to transfer them responsibility would take the form of some kind of border-carbon-taxation or adjustment, which are usually more related to CBA evaluation proposals. We have provided some results and discussion on all this.

Comment: " the estimation of the penalty is based on the total amount of emissions related to international trade. Why do you not estimate ERA comparing differences (domestic - imported) by sectors averages? How would the responsibilities change?"

Answer:

We have computed the results by sectors (see e.g. Table D4 in the SI) and find that the largest difference between ERA and CBA for a sector (a penalty of 68 MtCO₂ for the sector "c14: Electrical and Optical Equipment, which ranked 3rd in total ERA value, see Figure 3) is about half of the country's overall penalty (which was 129 Mt CO₂).

We have also included the results of the bilateral trade between China and the US, which yields a penalty of 61 MtCO₂. This is driven by the penalties for "c14: Electrical and Optical Equipment" (the largest in absolute terms, explaining 35.7 MtCO₂), "c4: Textiles and Textile Products" (5.6 MtCO₂), "c13: Machinery, Nec" (5.1 MtCO₂) and "c16: Manufacturing, Nec" (4 MtCO₂).

Comment: Also in terms of equity, it could be also useful to analyze the increases or reductions of responsibilities after the application of the ERA criterion, compared to PBA and CBA, by a list of countries organize by per capita income. This would be helpful to know whom, clean rich or dirty emerging countries, are more benefited. Is there any relationship with the level of development?

Answer: we have analyzed the increases or reductions of responsibilities after the application of the ERA criterion, compared to PBA and CBA, by a list of countries organized by per capita income. We may observe that that the richest

countries (with a total credit of 35 MtCO₂) trade “better” (in general with higher credits) than the poorest countries (with a penalty of 39 MtCO₂). It is also clear that the results are dominated by the very large penalties for China (129 MtCO₂) and the US (72 MtCO₂). As a consequence, most remaining rich countries (all 20 except the US) still trade better than most remaining poor (all 20 except China), but the gap between rich and poor has nearly vanished (total credits of 107 and 90 MtCO₂) and almost all perform better than the average.

Comment: - MRIO vs BRIO: the use of the MRIO model to estimate ERA in front of CBA, makes countries responsible for their consumption (but not for production). Subsequent emissions reductions commitments have to be transferred to industries. In this line, you have to punish or credit industries, so it seems also interesting to compare production technologies more than consumption patterns. All those considerations bring me to consider the use of a Biregional input-output model (BRIO) instead of an MRIO model. Using this approach, you compare the domestic efficiency of the importer country, with the domestic efficiency of the exporter. This would be helpful also to provide better evidence about the industries at risk of carbon leakage to reformulate the forthcoming revisions of the EU-Emissions Trading Scheme. Have you considered the application of a BRIO model? Have you thought about the pros and cons of it?

Answer: We have considered the application of a BRIO model but we still consider that the MRIO model provides a better reflection of all the emissions involved across the global supply chains. We consider that the “efficiency” of an exporter and of an importer is better reflected by accounting for what happens globally with their bilateral trade, and this global change needs to capture the flows across all the economies, which is also valid for capturing carbon leakage.

The literature on aggregation biases suggests that the best approach is to run the calculations at the most detailed level (with as many as possible countries) and aggregate the results afterwards, when presenting the results. Working with an aggregated model or just a part of the full model (like the bi-regional model) implies that interregional spillover and feedback effects are not taken into account. In particular due to the enormous increase in trade in intermediate inputs (caused by international fragmentation) these spillover and feedback effects have become more important.

Comment: - Results: There are no results in section 4. In my opinion, although your paper could be considered as a theoretical approach, the empirical dimensions of the consequences of the ERA criterion have to be analyzed. I recommend you to provide a deep analysis of the ERA by countries, years, comparison with the rest of approaches, etc. And, of course, this has to be presented using more attractive figures, tables and diagrams, as I will recommend you in the next comment.

Answer: We have developed an in-depth analysis of the ERA by countries, years, comparison with the rest of approaches, etc. By far the largest outcome in size is the penalty (difference of ERA and CBA) of 61 MtCO₂ for the trade between the US and China. Between 1995 and 2009 trade among the two countries grew more than 6 times on a nominal basis. A trade war between

these countries, where one does not allow imports from the other and vice versa, would have at least been beneficial for the global CO₂ emissions in 2009. In terms of percentage change, it implies an increase of more than 700% increase in the value of CBA of Chinese responsibility of CO₂ emissions of imports from the US. Other important increases in penalties are those between China and RoW.

Comment: - Results visualization: the results visualization is very poor. It is in line with the scarcity of the results analysis. I consider that better and more readable figures and tables are needed. Please, use better tools, colors, change the thickness of lines, rescale axis, etc. to improve the readability and to identify different patterns, behaviors, trends, etc. Consider the possibility to change the Table 1 (or place it in the Supporting Information) and use another type of graphic to illustrate the real differences between criteria and countries (who is more credited? who is the most penalized?). In this sense, the use of a kind of Sankey diagram or CIRCOS graphic presenting figures related to the size of penalties and the origin of the most penalized trade relationship could be useful in order to know who are the winners and losers of ERA (I follow the recommendations suggested in previous comments).

Answer: Following the reviewer's suggestions, we have developed some figures, especially to highlight who would have been the winners and losers with credits/penalties from 1995-2009 according to ERA. Additional Figures are given in the Online Supplementary Information.

Comment: - Desirable properties, section 5: the discussion of the properties is always interesting, as is possible to read in Kander et al. (2015), Domingos et al. (2016) and reply. That is why I consider necessary its inclusion in this paper. However, section 5, after results, is not the best place to insert it. In my opinion, it should be placed in section 3 where the ERA is proposed.

Answer: We have placed this discussion of the properties in section 3 where the ERA is proposed.

Comment: - Discussion and policy implications: I consider that this discussion must be focused in the adequation of the new proposal in the international context. Moreover, subsequent policy implications derived from the application of ERA criterion would be explained in detail. In my opinion, the description of the Kyoto Protocol mechanisms is not useful in this context, Kyoto is the past, and the justification of the alignment of ERA with the Paris Agreement tools (ITMOs) is not clear. How ERA helps the instrumentalization of ITMOs like SO₂ trading systems in the USA, the new EU-ETS (considering that all the countries should have their own cap-and-trade programs) or promotes Clean Development Mechanisms? The way you would implement it (via Carbon Taxation using mechanisms like those proposed in Springmann 2014?) should be discussed here, in comparison with the absence of alternative incentives following the PBA or CBA criteria. The results show by ERA are really close to CBA, so is hard to imagine why here's proposal is in line with the recommendations of Article 6 of the Paris Agreement, considering that PBA was

proposed as the allocation criterion. In fact, you propose this idea since the abstract, and I cannot identify a clear relationship. Maybe you could try to clarify the relationship.

Answer: We have tried to make a much stronger case on the relation between the ERA and the climate policy implications. To start with, we have minimized the description of the Kyoto Protocol mechanisms because they are just used to give further context to the general mitigation options.

We have tried to emphasize across the text the clear nature of ERA for indicating the right track and provide guidance and signals towards reducing emissions. The Paris Agreement requires documenting the evolution but does not establish clear guidance. We agree that ERA is not directly reflecting Article 6 of the Paris Agreement (stating that PBA was proposed as the allocation criterion). However, it certainly does more than PBA or CBA in this respect: That is, following the recommendations and offer possibilities to track the evolution and effectiveness of actions, and to provide consistency and guidance for measuring and orienting mechanisms of transferring mitigation outcomes. In this way, ERA results can be turned into indicators that point countries at actions that reduce global emissions (e.g. with a certain partner, with a bilateral agreement, etc.).

With respect to a possible next step in the direction of implementation of the ERA system, it should be noted that Article 6 of the Paris Agreement provides the opportunity to expand the reach of carbon pricing to enable full implementation of Nationally Determined Contributions (NDC). It is also emphasized that its development should be guided by the fundamental principles of ensuring environmental integrity and avoiding double counting (see e.g. IETA, 2016). Indeed, mechanisms like those proposed in Springmann (2014) seem a step forward, and add to the proposals of border-carbon-taxation or adjustment.

Again, thank you very much for the useful comments which we have followed, together with the comments by the other reviewers and the editor, while trying to keep the article within a reasonable length. We hope that this revised version meets the demand for a clearer exposition of the methods, numerical results and policy discussion.

Reviewer #2 (Remarks to the Author):

Answer: We thank the referee very much for his/her very useful comments. Below, we address point-by-point the suggestions and indicate the changes they have induced.

Comment: The article is in general well written, the points well argued, and the results and message relevant for the climate change community. The article is perhaps somewhat too much on the technical side for Nature Communications, but perhaps this is unavoidable given the nature of the issue at hand.

Answer: Thank you very much for the assessment. As described below, we have tried to provide a much more detailed discussion of the results and paid more attention to policy implications. This has shifted the focus from the technical aspects and to the issues of climate policy. Most technical details are now in the Online Supplementary Information.

Comment: One main concern is that the significance of the work doesn't really jump at you. In the introduction, the reader must get a feeling not only for why this may be new and different to what has been done before, but also for why this is important? What is actually at stake? Differences of hundreds of megatonnes depending on how you account? Negotiation strategies depending on this? Etc etc.

Answer: Yes, we had to stress much more the importance of the article, the method, and the implications. In particular in the introduction we have now highlighted why this is important, and what is at stake. The accounting of ERA provides the right incentives to countries to make efforts towards global reductions of emissions, complying with some desirable properties indicated previously in the literature.

The ERA method provides a reliable and consistent background for aspects related to policy implications. To start with, we see a very important role for ERA as a tool to track and document the progress of individual and collective contributions. Measurement, reporting and verification (MRV) of greenhouse gas mitigation is key in climate action. This not only holds for the emission inventories but also for mitigation actions and support. It is planned that the first full global stocktaking will take place in 2023 (UNFCCC, 2015, Decision paragraph 20 and Article 14) and will occur every five years thereafter. To assist countries in the implementation of their actions, a facilitative implementation committee of experts has been agreed upon (UNFCCC, 2015, Article 15). However, the exact modalities of the review remain to be determined in the years to come (Röser, Fekete, Höhne, & Kuramochi, 2015). As we have highlighted in our paper, ERA is a system which reveals global emission changes. It reflects any action regarding production and consumption and provides credits/penalties for actions on the trade between any two countries. As indicated in the Agreement, any comprehensive assessment should look into the global effects of reduction of emissions, and it is known that national actions may not be compatible with the agreed long-term goals of the Paris Agreement (see Röser et al., 2015). The Paris Agreement foresees a process that evaluates: the progress of individual Parties in meeting their NDCs,

and the overall accumulated progress in avoiding dangerous climate change. Quantification of the changes in global emissions will direct the policy makers back to the underlying actions that were taken. This will yield *ex-ante* suggestions for better actions in the future as guidance to the policy makers. Such suggestions may come from ERA findings. Ultimately, these can be turned into indicators on what countries should do in terms of choosing/prioritizing actions which lead to larger global emissions reductions.

Secondly, Article 6 of the Paris Agreement provides the opportunity to expand the reach of carbon pricing to enable full implementation of NDCs and to use Internationally Transferred Mitigation Outcomes (ITMOs). However, it is also emphasized that their evolution should be guided by the fundamental principles of ensuring environmental integrity and avoiding double counting (see e.g. CCAP, 2016, IETA, 2016).

A framework like ERA may help governments and international institutions account for emission reductions achieved. With national frameworks chances are small that countries can do this without double counting or other inconsistencies (each country will probably try to take the figure which is of interest and this may have been accounted elsewhere), or without a clear link between the effort and the impact (accounting for a benefit in emissions to which the country did not contribute). We think ERA is an important step forward in overcoming these issues.

Comment: The authors mention (in the intro) Gallego's shared responsibility approach (using eg value added shares as responsibility weights), but then IO cannot see this taken up anywhere in the analysis (e.g. at the outset of section 2). Why not? Already the sentence that follows the mention continues dealing with PBA and CBA only.

Answer: Gallego and Lenzen's article is important in the literature that focuses on the attribution of responsibilities. Our paper follows the literature that tries to solve challenges of complying with the cited properties (Kander et al. 2015 and related articles) and formulate three "important and intuitively compelling" properties for a scheme of credits and penalties.

Comment: "the properties of "sensitivity", "monotonicity" and "additivity" " need to be explained for the lesser prepared reader – this is not straightforward. I know this comes in section 5, but it should be motivated at least to not leave the reader hanging.

Answer: We have explained these properties in the introduction in a simple form, while also anticipates the discussion which has now been moved to Section 3.

Comment: I understand that the mathematical content of section 2 is needed. However, for the more general readership of Nature Communications, I would farm this out to an Appendix and present the same content in a more accessible form. This may be challenging, but would benefit the article if done well. The same applies to Section 3.1.

Answer: We have done our very best to reduce the technical/mathematical aspects in the main text, relegating the full explanation to the Online Supplementary Information. We have tried to make the exposition more intuitive and accessible for a general readership of Nature Communications. Yet, some equations are necessary to fully grab the contents of the method.

Comment: The authors say that the symmetry criterion is not relevant in an assessment confined to an upstream perspective (if I understood this right), but they could perhaps mention that an extension to a downstream perspective is straightforward, and already laid out by Gallego and also in Lenzen, M. & Murray, J. Conceptualising environmental responsibility. *Ecological Economics* 70, 261-270 (2010).

Answer: We have cited the comprehensive frameworks on upstream and downstream perspectives laid out by Gallego and Lenzen (2005) and Lenzen and Murray (2010).

Comment: Minor comments:

- typo: "should credit actions that contribute to reduced global emissions AND SHOULD penalize actions that increase them"
- This is because foreign emissions are embodied in the intermediate inputs – in an MRIO framework, aren't there also direct imports into final demand that carried emissions embodiments, since y is a matrix?
- "We argue that emission accounting is one thing (as PBA and CBA do)"- the grammar in that sentence is not right.

Answer: Ok, thank you for pointing these out, we have revised according to the suggestions.

Again, thank you very much for the useful comments which we have followed, together with the comments by the other reviewers and the editor, while trying to keep the article within a reasonable length. We hope that this revised version meets the demand for a clearer exposition of the methods, numerical results and policy discussion.

Reviewer #3 (Remarks to the Author):

Answer: We thank very much the referee for his/her very useful comments. In the following lines, we explain point by point the changes introduced according to the suggestions.

Comment: The manuscript by Erik Dietzenbacher et al. „Towards a more effective climate policy on international trade“ proposes a scheme for assigning responsibilities for emissions among countries accounting for trade interdependencies. In particular, the scheme credits actions that reduce global emissions and penalize actions that increase them. The manuscript take up a discussion that has been taking place in Nature Climate Change (Kander et al., 2015; Domingos et al., 2016) and, in fact, provides important clarifications and corrections. The manuscript is well written and I could not find problems in the underlying math. I still reject this manuscript, because I do not perceive it to be of interest to a broad readership as served by Nature Communications. It should be submitted to a more specialized journal (which, in fact, would also better suit the current structure of the manuscript).

Having said that, I would like to qualify that I would equally have rejected the pieces by Kander et al. (2015) and Domingos et al. (2016) on the same grounds (despite their additional problems). I provide this qualification for transparency to enable the editors to over-rule my judgement in case of disagreement.

Answer: We appreciate the sincerity and thank the positive comments which are also provided. We have tried to make a much stronger case on the relation between the ERA and the climate policy implications, reducing the technical side of the article and relegating the details to the Online Supplementary Information, and touching more on the analyses, discussion of policy implications.

Comment: 1) Relevance of consumption-based accounting in the context of the broader discussions on responsibility in international climate policy: There is a wider discussion on assigning responsibility for greenhouse gas emissions and anthropogenic climate change that goes far beyond consumer and producer responsibility. Accounting for trade in the assignment of responsibility is usually motivated in terms of developing countries as their producer emissions are much higher than their consumer emissions. Yet, the difference between consumer and producer responsibility is dwarfed by extending the relevant time frame back to pre-industrials as proposed in the discussion around historical responsibility. Of course, one discussion does not exclude the other – particularly, given the fact that, in principle, historical trade-adjusted emissions accounts could be developed (even though this is a theoretical argument as constructing reliable accounts seem impossible). Yet, trade-adjusted accounts have remained a small academic discussion and not received much attention in international climate policy, while historical responsibility has been a key concern in the negotiations for many developing nations from the beginning. In my view this will not change in the future – partly due to the point I will explain next.

Answer: We agree with the reviewer that there is a wider discussion on assigning responsibility that goes far beyond consumer and producer responsibility, e.g. the discussion around historical responsibilities. Similarly, we are also aware of the obstacles for changing some views, inertias, etc. However, fact is that 20-30% of global emissions are in the exports. If we can reduce these emissions by stimulating –and indicating how– to trade in a ‘smarter’/‘cleaner’ way, focus on (also) trade seems justified.

Next, we follow the reviewer and assume that historical trade-adjusted emission accounts *can* be developed. We agree with the reviewer that “this is a theoretical argument as constructing reliable accounts seem impossible”. In that hypothetical case, the ERA approach could be used to adapt the trade-adjusted accounts. We might, for example, observe that, in a world with mostly closed economies, early trading nations might be heavily penalized/credited for their trade. It should be stressed that a lot of actions to reduce emissions are already covered by PBA and CBA measures. ERA adapts CBA by focusing on extra gains/losses due to trade.

In international negotiations on the assignment of burdens/efforts/responsibilities, aspects related to the historical responsibility and the capacity to pay will remain to dominate the initial setting. Although these aspects have not been at the heart of the Nationally Determined Contributions (NDCs), they are implicit in the discussions. Several authors see implementation and strengthening of the Paris Agreement as the next step for the global response to climate change (see e.g. Höhne et al., 2017.) So, once these aspects for the initial setting have been established, one may add the ERA framework in future revisions of the Paris Agreement (e.g. as a tool for informing the Global Stocktake).

In our paper, we have tried to make clear and highlight the usefulness of the ERA method. It provides a reliable and consistent background for aspects related to policy implications. To start with, we see a very important role for ERA as a tool to track and document the progress of individual and collective contributions. Measurement, reporting and verification (MRV) of greenhouse gas mitigation is key in climate action. This not only holds for the emission inventories but also for mitigation actions and support. It is planned that the first full global stocktaking will take place in 2023 (UNFCCC, 2015, Decision paragraph 20 and Article 14) and will occur every five years thereafter. To assist countries in the implementation of their actions, a facilitative implementation committee of experts has been agreed upon (UNFCCC, 2015, Article 15). However, the exact modalities of the review remain to be determined in the years to come (Röser, Fekete, Höhne, & Kuramochi, 2015). As we have highlighted in our paper, ERA is a system which reveals global emission changes. It reflects any action regarding production and consumption and provides credits/penalties for actions on the trade between any two countries. As indicated in the Agreement, any comprehensive assessment should look into the global effects of reduction of emissions, and it is known that national actions may not be compatible with the agreed long-term goals of the Paris Agreement (see Röser et al., 2015). The Paris Agreement foresees a process that evaluates: the progress of individual Parties in meeting their NDCs, and the overall accumulated progress in avoiding dangerous climate change. Quantification of the changes in global emissions will direct the policy makers back to the underlying actions that were taken. This will yield *ex-ante*

suggestions for better actions in the future as guidance to the policy makers. Such suggestions may come from ERA findings. Ultimately, these can be turned into indicators on what countries should do in terms of choosing/prioritizing actions which lead to larger global emissions reductions.

Secondly, Article 6 of the Paris Agreement provides the opportunity to expand the reach of carbon pricing to enable full implementation of NDCs and to use Internationally Transferred Mitigation Outcomes (ITMOs). However, it is also emphasized that their evolution should be guided by the fundamental principles of ensuring environmental integrity and avoiding double counting (see e.g. CCAP, 2016, IETA, 2016).

A framework like ERA may help governments account for emission reductions achieved. With national frameworks chances are small that countries can do this without double counting or other inconsistencies (each country will probably try to take the figure which is of interest and this may have been accounted elsewhere), or without a clear link between the effort and the impact (accounting for a benefit in emissions to which the country did not contribute). We think ERA is an important step forward in overcoming these issues.

Comment: 2) Input-Output Analysis as basis for certified and verified emissions reductions: For me one fundamental question in this entire discussion is whether we believe that political consensus could be achieved on such a scheme and its methodological and data foundation. Put in a different way, do we expect that the proposed scheme could be the basis for certifiable and verifiable emission credits or penalties? I think this is far from perceivable. Trade-adjusted accounts required to make the necessary calculations rely on input-output analysis. The various error sources have been discussed extensively in the scientific literature (e.g. Lenzen et al., XXX). The authors themselves, for example, point towards the fact that aggregated nature of input-output tables make it impossible for the method to be scale-invariant. In general, aggregation bias highlights that the level of aggregation of the input-output calculus partially determine the results. This makes the tool impractical as a basis for verified emission credits or penalties. While input-output tables work usually at the level of dozens or at best a few hundreds of sectors there are ten-thousands of products that are internationally traded. In the absence of reliable product level accounts in the foreseeable future, I perceive this as a merely academic discussion that can influence climate policy at best on a rhetorical level.

Answer: The reviewer makes a very thoughtful and sensible comment. In order to achieve a generally accepted scheme which “credits/ penalizes” in the way we described, global multiregional input-output (MRIO) data are necessary. All this is true, of course, but recall that IO has made enormous steps forward in the last ten years. The first publication using a truly global MRIO table was only in 2009 (Hertwich and Peters, 2009). The critique on using IO techniques was even much stronger before 2009, and was a trigger for several research teams to start building global MRIO tables. Today, the work on data issues continues and progress is continuously made. An important step forward was the fact that also a supranational institution like the OECD started to produce their global MRIO tables.

That being said, it is true that IO data will always contain some aggregation and estimation. The outcomes of any IO analysis will therefore contain biases. Yet, we would like to provide three arguments why IO is still very useful in the present context.

First, IO can start an academic discussion that can influence climate policy on a rhetorical level, and we think this is already important. The possibility of considering the role of trade and the fact that countries can mitigate by taking actions that affect foreign emissions was not even taken into account two decades ago. Now, it is part of the debate and has even been given a role to achieve emissions reductions (for example, through the Internationally Transferred Mitigation Outcomes, ITMOs). Also, an IO-based analysis would be able to quantitatively compare scenarios. For example, scenario A for the reduction in emissions due to the installation of solar panels in the EU, and scenario B for reduction in emissions if the same amount of money had been invested in China or India to change some technology or modes of production for lower-intensity ones. In this line, Steiner et al. (2014) argued that a consumption-based climate policy approach by an industrialized country needs to achieve a just and cost-effective support to emerging economies, in the form of financial and technological transfers. In the agreement, developed Parties shall provide information on the support provided to developing Parties (financial and technology transfers and capacity building). Developing Parties shall provide information on the support needed and received. The evaluation of the effectiveness on global emissions of such well documented actions then can be a very interesting exercise for academic (and we claim also for policy) discussions.

Second, moving from the academic side to the practicalities of the Paris agreement, we see a very important role for ERA as a tool to support Measurement, reporting and verification (MRV) of greenhouse gas mitigation. Results from IO models can be very useful to that end.

Third, we agree on the challenges related to data gathering and aggregation issues. At the same time, however, many voices call for a clearer framework regarding: the design of future NDCs (e.g. Criqui et al., 2018); the climate finance discussions of the NDCs (e.g. USAID, 2016); and notably the tracking of the progress of individual and collective contributions (e.g. Peters et al., 2017, Fyson et al., 2017). Also, much progress has already been made in combatting the challenges. As we argue in the final paragraphs of the revised version of our paper, more and more datasets have become available in the last decade. Such databases are now gathered and updated timely and in a consistent manner. A key step would be to decide on which global MRIO database to use, although results from database-comparison exercises would be also useful (as in the model intercomparison exercises of the IPCC). Once that is done we will have a point of reference and the discussions on scale will become obsolete. It should also be borne in mind that such discussions have already lost much of their relevance. Moran and Wood (2014) have shown that the main results for carbon footprint accounting converge across databases. Moreover, the discrepancies are basically due to using different definitions (and data) for the environmental stressors (i.e. PBA data) and not much due to differences in (the data for) the economic structure, trade, consumption (i.e. global MRIO data).

Comment: 3) Structure of the manuscript: The manuscript with its focus on methodological details is written for a field journal rather than a broader readership. Of course, this could be changed. If accepted, this needs to be changed. Most of the details of sections 2 and 3 should be shifted to the Supplementary Information. Results could be discussed in more depth. The discussion would need to be extended to discuss relevant limitation of the proposed method and link this to requirements for application in political processes like the UNFCCC.

Answer: We have done our very best to reduce the technical/mathematical aspects in the main text, relegating the full explanation to the Online Supplementary Information. We have tried to make the exposition more intuitive and accessible for a general readership of Nature Communications. Yet, some equations are necessary to fully grab the contents of the method. Most of the details of sections 2 and 3 are now shifted to the Online Supplementary Information. The discussion has been extended to discuss relevant limitations of the proposed method and linked to requirements for application in political processes like the UNFCCC.

Comment: Some specific comments on the manuscript: Just a few additional comments.

- As highlighted before a discussion of the limitations of input-output analysis for emission accounting is missing (various biases).
- Figures all lack axis labels
- Data description in my view is insufficient. We have, for example, no idea what CO₂ emission component this is – as it is obviously not all anthropogenic CO₂. How does it compare to standard accounts (IEA, BP, EDGAR etc.)?
- No equation numbers make things cumbersome.
- Equation on page 7: It is entirely unclear why changes in emissions should be equally shared by countries. See Gallego and Lenzen (2005).

Answer: OK. We have revised all these aspects. We have:

- introduced axis labels,
- extended the data description,
- discussed the sharing of changes in emissions by countries,
- discussed the limitations of input-output analysis for emission accounting,
- provided the details of the satellite accounts from WIOD (some results only for CO₂ emissions, additional results for CO₂ equivalent emissions including CO₂, CH₄ and N₂O). As explained in Genty et al. (2012), the accounting framework for environmental variables that sets data up to be juxtaposed to the MRIO is consistent with input-output data (residence principle, in which emissions of a resident unit, no matter whether physically in or out of the territory, are allocated to the territory of residence). This is different from (but can be linked to, see Genty et al., especially Table 2) the framework of energy balances such as those of the IEA, or EDGAR, which allocate all emissions of economic units to the country where they physically (geographically) take place, regardless of whether they are undertaken by residents or non-residents.

Again, thank you very much for the useful comments which we have followed (together with the comments by the other reviewers and the editor), while trying to keep the article within a reasonable length. We hope that this revised version meets the demand for a clearer exposition of the methods, numerical results and policy discussion.

Reviewers' comments:

Reviewer #1 (Remarks to the Author):

Dear authors,

Please find here the second round of comments to the authors about the proposed manuscript "Towards a more effective climate policy on international trade".

First, I would like to thank the authors for the insightful revision and answer of all my comments and queries. Although there are still some things we could discuss, I have to understand that both, the proposal and its justification, fit perfectly with the standards of a high-quality publication and represent the author's point of view of their contribution. From the side of the reviewers, we have to incentivize the correction of errors, but we cannot pretend to impose our personal points of view. Second, in my opinion, in the current's version, the paper fulfills all the requirements to be published in Nature Communications. The revision held has improved the insights of the proposal, and things related to results visualization are more refined now. In this sense, in my opinion, this paper contributes with novelties and improvements to the existing literature and opens and presents useful tools to reconsider the way we are accounting emissions responsibilities helping to overcome some of the typical problems when alternatives to the main criteria (producer vs. consumer-based approaches) are evaluated. That is why I would recommend this paper for publication in Nature Communications.

Anyway, despite these positive comments, I would like to reveal that the revision period took too much time, which could represent a minor problem. The delay is perfectly explained by the deepness of the reviewer's comments and queries, this does not suppose any insuperable inconvenience. Nevertheless, "many things" have happened and have been proposed in the existing literature in the last months that should be considered in this paper. In my opinion, to sum up, there are three recent publications in Nature Communications that could help to improve the proposal and could even strengthen it. I would like to recommend the reading of them, as well as I would like to recommend that a reflection be included in how they may or may not affect your proposal. The papers and my reflection suggestions are:

- Chen Z-M, Ohshita S, Lenzen M, Wiedmann T, Jiborn M, Chen B, et al. Consumption-based greenhouse gas emissions accounting with capital stock change highlights dynamics of fast-developing countries. Nature Communications. 2018; 9:3581.

How can the proposal presented in this paper affect your ERA proposal? The dynamization of the models endogenizing capital, taking into consideration the contributions of different times and different regions capital inputs, would result in different CBA results compared to the traditional static vision of capital in MRIO models.

If we assume that capital goods and consumption goods could be treated differently, maybe your ERA's proposal is not valid for capital goods. E.g., in the case of photovoltaics (PV) solar panels, China

is leading (by far) the world's manufacture process. Could the penalties imposed on China slow down the world's energy transition by installing low emissions technologies, such as the PV solar panels? Could you raise some reflections regarding this and the ERA proposal estimations?

- Meng J, Mi Z, Guan D, Li J, Tao S, Li Y, et al. The rise of South–South trade and its effect on global CO2 emissions. *Nature Communications*. 2018; 9:1871.

The ERA proposal fits perfectly in a global framework of developed and developing countries generating adequate incentives (in your opinion) derived from the credits or penalties that result from the application of the proposal. However, the evidence shown in the suggested to read paper shows an increase in emissions intensities between emerging and less-developed regions, that it could be the case of China and Vietnam. I think that includes this evidence in your work and assess how the ERA's results might be affected could improve the quality of your paper.

- López L-A, Cadarso M-Á, Zafrilla J, Arce G. The carbon footprint of the U.S. multinationals' foreign affiliates. *Nature Communications*. 2019; 10:1672.

Multinational enterprises represent one of the drivers of technology transfers between the multinational parent's country and hosting's countries. Could the new approach of estimating the multinational enterprises' carbon footprint presented in this paper be related to your contribution? In an increasingly dominated world by big data, day by day we are getting closer to firms. In fact, the forthcoming ICIO-OECD database will provide detailed information about the activity of multinationals firms and processing exports in emerging economies. Could this represent future developments regarding the ERA approach? Under the assumptions of your proposal, how could multinational parent's country transfer the responsibility to firms?

Lastly, I am not sure if some of the decisions regarding using the Supplementary Information to, e.g., extending the methodological explanations are allowed by the journal's guide for authors. They must be within the Methods section in the article. Please spend some time reviewing the guide for authors suggestions and ask editors about them.

Thanks again, and congratulations for your interesting proposal.

Reviewer #2 (Remarks to the Author):

- The authors have improved the manuscript a lot, and with the technicalities now being outsourced, this is starting to look more like a NatComms paper. However, as I said in my previous review, the innovation and novelty doesn't "jump" at the reader; one has to search and find. I will explain more below.

- Despite now most of the maths being in the Appendix, it takes from line 190 to line 280 to get to the final ERA equation 7. Of course, from an academic's point of view, reading the manuscript, it seems necessary, and also the motivation that the authors provide is OK to follow. Still, compared with PBA and CBA, this is complicated. And because it's complicated it may be difficult to motivate decision-makers to take it up. Of course, complicated things (the transistor etc) have been adopted in practice, but this is to do with people's perception of what is fair, and here, people are unlikely to follow a mathematical equation, no matter how invariant monotonous additive symmetric etc etc. From experience, it takes at least five to ten minutes to explain CBA to a non-expert. But this ERA must take much much longer, thus decreasing its chances for becoming popular. Of course, an article should not not be published because the measure may not become popular. But the complicated nature of ERA may well impinge on its adoption by decision-makers. Could this be different? Maybe this is worth an idea in the discussion that is more reassuring than "Although it is not very likely that ERAs or a similar system of credits and penalties will be implemented in the short run, it certainly seems a viable option for the future"?

- The point above has implications for the manuscript flow. Because ERA is complicated it cannot be explained in one or two sentences. And therefore, the abstract does actually not explain what exactly ERA is. The paragraph starting in line 88 with "we argue that..." attempts such an explanation, but even after 10 lines one only know "sort of" what ERA is. With this in mind, the text from line 111 ("ERA is powerful..." "Paris Agreement..." etc) evokes some discomfort, because these feel like a mere assertions when one has not fully grasped what ERA actually is. Then follows lines 190 to 280, and essentially one has to understand ERA from an equation. Right to the end, I only had half an intuitive understanding what ERA is. I understand this might be challenging, but could it be said without misunderstanding, metaphor or glossing-over, what ERA is, in two sentences? This being right upfront in the abstract would really do it for me, because it would prepare me for believing what I read in lines 88ff, and would avoid me having to wait until line 280 to understand finally what this is about.

- In the discussion the authors say "In the discussion on (shared) producer and consumer responsibilities (which was briefly cited in the introduction), many authors have emphasized that both perspectives do with their focus on reducing global emissions and providing an incentive to consumers and producers to contribute through "smarter" trade." I understand that shared responsibility produces outcomes between PBA and CBA. In practical terms (let's forget for a moment theoretical underpinnings and properties), what is so different about ERA, compared to shared responsibility? What does ERA have that shared responsibility doesn't. That is still not explained sufficiently. I understand ERA is different to PBA and CBA, but this is kind of obvious, and explained well (perhaps in too detailed a way). Like ERA, shared responsibility also intended to improve on the two polar views. So we need to know why ERA is better than shared responsibility (if it is).

Reviewer #3 (Remarks to the Author):

I would like to congratulate the authors for a much improved manuscript. In particular, the fundamentally reworked result section and the moving of information to the supplementary material benefits the readability and appeal of the manuscript. Yet, I remain with my fundamental concerns that this is not interesting for the wider readership of Nature Communications. Despite all progress in input-output accounting, I do not share the perception that this COULD and SHOULD actually be used for climate policy-making due to the various sources of uncertainty involved (I provide more detailed explanations in my previous reviews - point 2). Therefore, from my point a very good manuscript that makes important conceptual point, which would be well targeted to a more specific field audience.

Reviewers' comments:

Reviewer #1 (Remarks to the Author):

Comment: -

Dear authors,

Please find here the second round of comments to the authors about the proposed manuscript "Towards a more effective climate policy on international trade".

First, I would like to thank the authors for the insightful revision and answer of all my comments and queries. Although there are still some things we could discuss, I have to understand that both, the proposal and its justification, fit perfectly with the standards of a high-quality publication and represent the author's point of view of their contribution. From the side of the reviewers, we have to incentivize the correction of errors, but we cannot pretend to impose our personal points of view. Second, in my opinion, in the current's version, the paper fulfills all the requirements to be published in Nature Communications. The revision held has improved the insights of the proposal, and things related to results visualization are more refined now. In this sense, in my opinion, this paper contributes with novelties and improvements to the existing literature and opens and presents useful tools to reconsider the way we are accounting emissions responsibilities helping to overcome some of the typical problems when alternatives to the main criteria (producer vs. consumer-based approaches) are evaluated. That is why I would recommend this paper for publication in Nature Communications.

Answer: Thank you very much for the positive comments.

Anyway, despite these positive comments, I would like to reveal that the revision period took too much time, which could represent a minor problem. The delay is perfectly explained by the deepness of the reviewer's comments and queries, this does not suppose any insuperable inconvenience. Nevertheless, "many things" has happened and has been proposed in the existing literature in the last months that should be considered in this paper. In my opinion, to sum up, there are three recent publications in Nature Communications that could help to improve the proposal and could even strengthen it. I would like to recommend the reading of them, as well as I would like to recommend that a reflection be included in how they may or may not affect your proposal. The papers and my reflection suggestions are:

Answer: Thank you for pointing out these new publications; we have reviewed them, together with a few other recent ones in Nature Communications or similar outlets and on similar topics. We have briefly discussed them at the end of the manuscript when pointing out future work. Detailed answers to the comments are given below.

Comment: - Chen Z-M, Ohshita S, Lenzen M, Wiedmann T, Jiborn M, Chen B, et al. Consumption-based greenhouse gas emissions accounting with capital stock change highlights dynamics of fast-developing countries. Nature Communications. 2018; 9:3581. How can the proposal presented in this paper affect your ERA proposal? The dynamization of the models endogenizing capital, taking into consideration the contributions of different times and different regions capital inputs, would result in different CBA results compared to the traditional static vision of capital in MRIO models.

If we assume that capital goods and consumption goods could be treated differently, maybe your ERA's proposal is not valid for capital goods. E.g., in the case of photovoltaics (PV) solar panels, China is leading (by far) the world's manufacture process. Could the penalties imposed on China slow down the world's energy transition by installing low emissions technologies, such as the PV solar panels? Could you raise some reflections regarding this and the ERA proposal estimations?

Answer: Thank you. Certainly, the move from a static to a dynamic model would change the results, as shown in Chen et al. (2018), but that does not mean that ERA's proposal is not valid for capital goods. Capital typically has widespread forms of being registered in national accounting, so that investments are reflected in the final demand, being the value of capital in the form of property and business rents and depreciation or consumption of fixed capital registered in the value added.

As a matter of fact, the introduction of investments doesn't change the principles underlying ERAs. Gross fixed capital formation implies the production of capital goods (just as consumption implies production of consumption goods). The capital goods are necessary for production (and thus indirectly consumption in the future). The only difference is in the timing of using the capital goods. In the case of consumption goods, they are produced (as are the inputs and the inputs into the inputs etc.) in the same year as they are consumed. Therefore, the responsibility can be assigned to this year's consumption. In the case of capital goods, the responsibility needs to be divided over a number of years and should be assigned to future consumption. Under certain assumptions, however, it is possible to view the production of capital goods in a year as reflecting the necessary investments to maintain the capital services. So, the principles remain the same for all final products (whether used for consumption or for investments). Of course, the numerical results may differ (depending on aspects of timing, which is related to the growth or decline of the economy).

In the proposed example of photovoltaics (PV) solar panels, it should be highlighted that if (as it seems) the manufacture process of them is not particularly energy intensive (based on the inputs, etc.), the sector (if available separately or the general sector of "machinery & equipment" to which PV belongs) should not be particularly energy intensive, and hence not differ much from the trading partners and hence have similar penalties from bilateral trade. Still if it is the case, pure economic logic would tell that if manufacturing such a good in China is discouraged, it will be produced somewhere else (namely in a country with a "comparative advantage" after internalizing the penalty for emitting).

On the other hand, production does not occur just for export. In terms of accounting, investment in PV in China (not for export) are an increase in the gross capital formation (final demand), which apart from modifying the share of value added for the sector itself, it ultimately also changes the technology (the column structure of the IO) of other goods and services, when used to supply them electricity from PV, reducing the emissions and hence reducing the potential penalties (or converting them into/increasing credits) from exports of those goods and services. In that regard, ERA does not discourage by itself manufacturing certain goods at certain places (emission friendly or not) but it does so in as much as changes in bilateral trade help reducing global emissions.

Comment: - Meng J, Mi Z, Guan D, Li J, Tao S, Li Y, et al. The rise of South–South trade and its effect on global CO2 emissions. Nature Communications. 2018; 9:1871.

The ERA proposal fits perfectly in a global framework of developed and developing countries generating adequate incentives (in your opinion) derived from the credits or penalties that result from the application of the proposal. However, the evidence shown in the suggested to read paper shows an increase in emissions intensities between emerging and less-developed regions, that it could be the case of China and Vietnam. I think that includes this evidence in your work and assess how the ERA's results might be affected could improve the quality of your paper.

Answer: There is certainly evidence that trade among developing nations (i.e., South–South trade) has increased recently (more than doubled between 2004 and 2011) and that complex supply chains are distributing energy-intensive industries and their CO₂ emissions throughout the global South. We have briefly mentioned this, and certainly with more updated tables we will be probably seeing how credits/penalties will reflect the moves of production (especially of energy intensive activities) across these countries (seemingly slowdown in China, while increase in India, Vietnam or Pakistan), and whether that trade creates an issue for global emissions. For example, with the current technologies it seems that China-India trade does not imply much credits/penalties, in the sense that (embodied) emission intensities are quite similar among the regions. It should be stressed that the methodology of ERAs is not affected. It is true though that using a more disaggregated database in terms of products (such as EXIOBASE, or GTAP) and regions (GTAP or Eora) sheds more light on specific environmentally relevant industries or reflects better the details of “South-South” trade.

Comment: - López L-A, Cadarso M-Á, Zafrilla J, Arce G. The carbon footprint of the U.S. multinationals' foreign affiliates. *Nature Communications*. 2019; 10:1672.

Multinational enterprises represent one of the drivers of technology transfers between the multinational parent's country and hosting's countries. Could the new approach of estimating the multinational enterprises' carbon footprint presented in this paper be related to your contribution? In an increasingly dominated world by big data, day by day we are getting closer to firms. In fact, the forthcoming ICIO-OECD database will provide detailed information about the activity of multinationals firms and processing exports in emerging economies. Could this represent future developments regarding the ERA approach? Under the assumptions of your proposal, how could multinational parent's country transfer the responsibility to firms?

Answer: The new approach of López et al. (2019) and the main issues around multinationals could be related to the contribution, in our view especially in better defining where the value added and responsibilities are. The (possibly) forthcoming ICIO-OECD database is said to include information on the activity of multinational firms and processing exports. Also other initiatives are employed to further disaggregate existing input-output tables. For example, distinguishing production by small/medium/large firms, production in cities, distinguishing between urban and rural areas or between sexes. Each of these extensions would certainly be useful and benefit our analyses by better distinguishing the existing heterogeneities (not only of regions and sectors, but also of enterprises). However, the ERA approach would not change. It may well obtain the credits and penalties from modified versions of the consumer-based approach that considers MNE foreign affiliates' responsibility, for example.

Comment: - Lastly, I am not sure if some of the decisions regarding using the Supplementary Information to, e.g., extending the methodological explanations are allowed by the journal's guide for authors. They must be within the Methods section in the article. Please spend some time reviewing the guide for authors suggestions and ask editors about them.

Thanks again, and congratulations for your interesting proposal.

Answer: Thank you; we did not find specific limitations on this in the guide for authors. Of course, we will follow any editorial suggestion or request in this regard. Thank you again.

Reviewer #2 (Remarks to the Author):

Comment: - The authors have improved the manuscript a lot, and with the technicalities now being outsourced, this is starting to look more like a NatComms paper. However, as I said in my previous review, the innovation and novelty doesn't "jump" at the reader; one has to search and find. I will explain more below.

Answer: Thank you; we have tried to make it "jump" more at the reader.

Comment: - Despite now most of the maths being in the Appendix, it takes from line 190 to line 280 to get to the final ERA equation 7. Of course, from an academic's point of view, reading the manuscript, it seems necessary, and also the motivation that the authors provide is OK to follow. Still, compared with PBA and CBA, this is complicated. And because it's complicated it may be difficult to motivate decision-makers to take it up. Of course, complicated things (the transistor etc) have been adopted in practice, but this is to do with people's perception of what is fair, and here, people are unlikely to follow a mathematical equation, no matter how invariant monotonous additive symmetric etc etc. From experience, it takes at least five to ten minutes to explain CBA to a non-expert. But this ERA must take much much longer, thus decreasing its chances for becoming popular. Of course, an article should not not be published because the measure may not become popular. But the complicated nature of ERA may well impinge on its adoption by decision-makers. Could this be different? Maybe this is worth an idea in the discussion that is more reassuring than "Although it is not very likely that ERAs or a similar system of credits and penalties will be implemented in the short run, it certainly seems a viable option for the future"?

Answer: . We have tried to indicate why the logic of ERA is not so difficult to explain to a non-expert. We have also tried to justify why ERA can truly be a viable option in the not so distant future, especially since there are interesting movements into the direction of generalizing accounting systems (e.g. OECD and WTO joining forces in their TiVA initiatives building Statistics on Trade in Value Added ¹). For example, we have modified the text in this vein in subsection "5.2. The potential of ERA for developing policy".

Comment: - The point above has implications for the manuscript flow. Because ERA is complicated it cannot be explained in one or two sentences. And therefore, the abstract does actually not explain what exactly ERA is. The paragraph starting in line 88 with “we argue that...” attempts such an explanation, but even after 10 lines one only know “sort of” what ERA is. With this in mind, the text from line 111 (“ERA is powerful...” “Paris Agreement...” etc) evokes some discomfort, because these feel like a mere assertions when one has not fully grasped what ERA actually is. Then follows lines 190 to 280, and essentially one has to understand ERA from an equation. Right to the end, I only had half an intuitive understanding what ERA is. I understand this might be challenging, but could it be said without misunderstanding, metaphor or glossing-over, what ERA is, in two sentences? This being right upfront in the abstract would really do it for me, because it would prepare me for believing what I read in lines 88ff, and would avoid me having to wait until line 280 to understand finally what this is about.

Answer: We have tried to explain ERA also to non-experts. The simplest way of telling “the story” is as follows. ERA adapts CBA-based responsibilities by giving additional credits and penalties. For a particular country is the size of the credit or penalty determined by how much CO2 emissions are saved globally due to the trade of this country (when compared to the savings achieved by the average country). This two-sentence explanation is upfront in the abstract, after which the idea is further developed later in the manuscript.

Comment: - In the discussion the authors say “In the discussion on (shared) producer and consumer responsibilities (which was briefly cited in the introduction), many authors have emphasized that both perspectives do with their focus on reducing global emissions and providing an incentive to consumers and producers to contribute through “smarter” trade.” I understand that shared responsibility produces outcomes between PBA and CBA. In practical terms (let’s forget for a moment theoretical underpinnings and properties), what is so different about ERA, compared to shared responsibility? What does ERA have that shared responsibility doesn't. That is still not explained sufficiently. I understand ERA is different to PBA and CBA, but this is kind of obvious, and explained well (perhaps in too detailed a way). Like ERA, shared responsibility also intended to improve on the two polar views. So we need to know why ERA is better than shared responsibility (if it is).

Answer: ERA should not be compared with or viewed as an alternative for shared responsibility. In a nutshell, PBA measures the emissions and CBA assigns them to final consumers, at home and abroad. Hence CBA involves trade in emissions. ERA adapts CBA by taking account different trading possibilities and incentivizing to trade ‘better’ or ‘smarter’. This implies that in the discussion on shared responsibility we propose to look at a mix of PBA and ERA, rather than a mix of PBA and CBA.

Note that it is *not* the case that ERA is between PBA and CBA (which is where the shared responsibility is). For example, Figure 1 shows that for the USA and for Brazil in the beginning of the period ERA is outside the range PBA-CBA.

Ultimately, the use of ERA avoids the trilemma whether to consider efforts in production, consumption, or trade. This is because ERA rewards actions towards global emissions reductions in any of these three areas.

Reviewer #3 (Remarks to the Author):

Comment: - I would like to congratulate the authors for a much improved manuscript. In particular, the fundamentally reworked result section and the moving of information to the supplementary material benefits the readability and appeal of the manuscript. Yet, I remain with my fundamental concerns that this is not interesting for the wider readership of Nature Communications. Despite all progress in input-output accounting, I do not share the perception that this COULD and SHOULD actually be used for climate policy-making due to the various sources of uncertainty involved (I provide more detailed explanations in my previous reviews - point 2). Therefore, from my point a very good manuscript that makes important conceptual point, which would be well targeted to a more specific field audience.

Answer: Thank you for the recognition of the improvements and for evaluating the paper “a very good manuscript that makes important conceptual point”. With respect to the remaining concerns of the reviewer, on the one hand we feel that we (and in our opinion also the editor) have different viewpoints regarding the potential interests of the (wider) readership of Nature Communications. Perhaps we should simply respect and accept these differences and not try to convince the reviewer (especially since we do not find a direct question or request by the reviewer). On the other hand, the editor explicitly asked to address the comments and we feel this discussion on uncertainty is interesting and relevant. Therefore, we try to comment on the possibilities using ERA for climate policy-making also in relation to the sources of uncertainty involved.

In our view "the various sources of uncertainty involved" are usually present in (basically) any model or approach for climate policy. Important for the IPCC reports have been the integrated assessment models (IAM-CGE², GCAM^{3,4}, IMAGE⁵, MESSAGE-GLOBIOM^{6,7}, REMIND-Magpie⁸, WITCH-GLOBIOM⁹, MERGE^{10,11}, EPPA¹², DICE/RICE¹³, ICAM¹⁴, MiniCAM¹⁵; see¹⁶ on how different models can yield considerably different results, and¹⁷ on the need of transparency for their usefulness). These models are used to study e.g. how 1.5°C or 2°C of global warming could be avoided at the lowest cost, in relation to countries' current pledges to cut emissions. The interrelations include the biophysical as well as the socioeconomic aspects, including functions on habits and preferences. These complex models involve many uncertainties and limitations. Yet, it is generally considered that they still offer valuable insights on how aspects such as demography, income, energy, land use and other resources systems, etc. interact among each other, and relate to climate challenge. In other words, these models (but also studies such as^{18*},¹⁹⁻²³) work under uncertainties. They are used because they are considered reasonable, or the best option available, or less wrong than tossing a coin (see reviews of contributions of IAMs providing relevant information to decision makers e.g. in^{13,24}).

In this regard, the question we pose is: can a system such as ERA, given the uncertainties it may have* (see below), be of use for climate policy? Or similarly, retaking an interesting question of the previous round of review made by Reviewer 3, do we expect that the

proposed scheme could be the basis for certifiable and verifiable emission credits or penalties? Our answers are positive, based on our reasoning below.

* We consider now Uncertainties for ERA:

- Input-output data gathering and modelling usual uncertainties: As discussed in works such as ²⁵⁻⁴⁷, and as pointed out in the previous round of reviews, error sources have been discussed extensively in the scientific literature. Based on the results from sensitivity analyses and knowing the key sources of uncertainty, researchers have evidence on the errors and their effects in specific cases and have been able to build up a good intuition.

- Levels of regional/sectoral disaggregation: Among the sources of uncertainty we could cite, it was pointed out that aggregation bias could be a serious one, and that the ERA computations will be hard to perform at the level of ten-thousands of products that are internationally traded. We agree, real-world detail is not present in much of the information (e.g. on environmental pressures) provided to citizens and institutions. Also, we do not think that building global input-output databases at the level of ten-thousands of products can be expected in the near future. On the other hand, product information and labelling can be done at the detailed level and the evaluation at the aggregate level with input-output tables. For many products it suffices to know the emissions in the inputs and the emissions in the inputs into the inputs. We think that this information exactly incentivizes the behaviour of consumers that is aimed at in the ERA approach.

- Choice of multiregional input-output database: Some researchers may argue that because of the existence of multiple accounting frameworks different answers will be provided to the same question. Agreement on the choice of the database (and for some questions also the method) may not occur. As we indicated before, e.g. Moran and Wood (2014) have shown that the main results for carbon footprint accounting converge across databases. If desired, one could average the results obtained from different input-output datasets. Furthermore, the more transparent these databases are, the easier it is to detect differences and underlying methodological choices, and ultimately arrive at convergence of results.

Another development that we observe is that different institutions join forces and are moving towards a single global database. A major step forward is that a supranational institution like the OECD started to produce global MRIO tables. In this regard, there are interesting movements into the direction of generalizing accounting systems, notably regarding trade accounting (e.g. OECD and WTO joining forces in their TiVA initiatives¹).

Once the discrepancies between databases have reduced sufficiently, another executive decision could simply be: pick one database and require that everyone runs computations with it for a certain period of time. Resources may be spend on research to increase the number of countries and/or industries (for which researchers rely on solid national data, not using crude extrapolations).

Although the previous option may seem very harsh, similar decisions have been taken before. For example, in the Kyoto Protocol most commitments were made whereas there were many uncertainties. Yet, countries were required to reduce their emissions of greenhouse gases below the 1990 levels. Were the estimates unquestionable or was there only one way of computing these levels? Certainly not. Another case was pointed out by Reviewer 3 in the previous round. Historical responsibility (extending the relevant timeframe back to the pre-industrial period) has been a key concern in the negotiations for many

developing nations from the beginning. Recent climate policy agreements have considered a ‘climate debt’[†] that wealthy countries owe to poor countries, as a result of their greater historical contribution to human-induced climate change. Clearly, ‘measuring’ emissions in the past century comes with much higher uncertainties than for example calculating ERAs. Nevertheless, gross numbers and intuitions clearly pointed out that Europe and the US emitted more than other countries in the past.

In the same fashion, ERA shows that some changes in trade (e.g. a switch in exports of high/low intensive sectors) between China-Europe clearly point at more/less emissions (and hence penalties/credits). For example, the current difference of ERA and CBA indicates that China-Europe trade increases global emissions (penalties), which holds for many products but especially for “c14: Electrical and Optical Equipment” and “c4: Textiles and Textile Products”. This is exactly the type of information that it is also needed for the Measurement, Reporting and Verification (MRV) of greenhouse gas mitigation. For MRV, a clearer framework is required and needs to be designed. This holds in particular for the tracking of the progress of individual and collective contributions (for which the Paris Agreement mentions ensuring environmental integrity, avoiding double counting, and Internationally Transferred Mitigation Outcomes).

A final remark is that one does not need to provide all information at the most detailed level of products or industries. What is important is to signal the right incentives. For example, import energy intensive products from countries that produce them with less emitting technologies. The consequences will clearly be reflected in ERA. As highlighted by ¹⁷ for IAMs, we emphasize the need of transparency about the database and accuracy related to aggregation levels. This goes in line with the calls for a clearer framework for tracking contributions to reduce emissions.

References of the Cover letter and Responses

1. OECD. Trade in value added. (2019). doi:<https://doi.org/10.1787/data-00648-en>
2. Fujimori, S., Hasegawa, T. & Masui, T. in (eds. Fujimori, S., Kainuma, M. & Masui, T.) 305–328 (Springer Singapore, 2017). doi:10.1007/978-981-10-3869-3_13
3. Edmonds, J. A. & Reily, J. M. *User’s Guide to the IEA/ORAU Long-Term Global Energy Economic Model with Carbon Dioxide Emissions: Personal Computer Version A84PC*. (1986).
4. Kim, S. H., Edmonds, J., Lurz, J., Smith, S. J. & Wise, M. The ObjECTS Framework for Integrated Assessment: Hybrid Modeling of Transportation. *Energy J. (Special Issue #2)* 51–80 (2006).
5. Stehfest, E. *Integrated assessment of global environmental change with IMAGE 3.0 : model description and policy applications*. (The Hague : PBL Netherlands Environmental Assessment Agency, 2014).
6. Krey, V. *et al. MESSAGE-GLOBIOM 1.0 Documentation*. (International Institute for Applied Systems Analysis (IIASA), 2016).
7. Fricko, O. *et al.* The marker quantification of the Shared Socioeconomic Pathway 2: A middle-of-the-road scenario for the 21st century. *Glob. Environ. Chang.* **42**, 251–267 (2017).

[†] Today climate-debt concept incorporates two distinct elements: 1) adaptation debt, which represents the compensation owed to the poor for the damages of climate change they have not caused. 2) emissions debt, which is compensation owed for their fair share of the atmospheric space they cannot use if climate change is to be stopped.

8. Marian Leimbach Lavinia Baumstark, Michael Luken and Ottmar Edenhofer, N. B. Technological Change and International Trade - Insights from REMIND-R. *Energy J.* **Volume 31**, (2010).
9. Bosetti, V., Carraro, C., Galeotti, M., Massetti, E. & Tavoni, M. WITCH A World Induced Technical Change Hybrid Model. *Energy J.* **27**, 13–37 (2006).
10. Manne, A., Mendelsohn, R. & Richels, R. MERGE: A model for evaluating regional and global effects of GHG reduction policies. *Energy Policy* **23**, 17–34 (1995).
11. Blanford, G., Merrick, J., Richels, R. & Rose, S. Trade-offs between mitigation costs and temperature change. *Clim. Change* **123**, 527–541 (2014).
12. Paltsev, S. *et al.* *The MIT Emissions Predication and Policy Analysis (EPPA) Model: Version4.* (2005).
13. Nordhaus, W. in *Handbook of Computable General Equilibrium Modeling SET, Vols. 1A and 1B* (eds. Dixon, P. B. & Jorgenson, D. W. B. T.-H. of C. G. E. M.) **1**, 1069–1131 (Elsevier, 2013).
14. Dowlatabadi, H. Integrated assessment models of climate change: An incomplete overview. *Energy Policy* **23**, 289–296 (1995).
15. Smith, S. J., Kim, S. H. & Pitcher, H. M. *Model Documentation for the MiniCAM.* (2003).
16. Wilkerson, J., Leibowicz, B., Diaz, D. & Weyant, J. *Comparison of Integrated Assessment Models: Carbon Price Impacts on U.S. Energy.* *Energy Policy* **76**, (2015).
17. Schneider, S. H. Integrated assessment modeling of global climate change: Transparent rational tool for policy making or opaque screen hiding value-laden assumptions? *Environ. Model. Assess.* **2**, 229–249 (1997).
18. Lawrence, M. G. *et al.* Evaluating climate geoengineering proposals in the context of the Paris Agreement temperature goals. *Nat. Commun.* **9**, 3734 (2018).
19. Fletcher, S., Lickley, M. & Strzepek, K. Learning about climate change uncertainty enables flexible water infrastructure planning. *Nat. Commun.* **10**, 1782 (2019).
20. Bordbar, M. H. *et al.* Uncertainty in near-term global surface warming linked to tropical Pacific climate variability. *Nat. Commun.* **10**, 1990 (2019).
21. Yumashev, D. *et al.* Climate policy implications of nonlinear decline of Arctic land permafrost and other cryosphere elements. *Nat. Commun.* **10**, 1900 (2019).
22. Scovronick, N. *et al.* The impact of human health co-benefits on evaluations of global climate policy. *Nat. Commun.* **10**, 2095 (2019).
23. Vandyck, T. *et al.* Air quality co-benefits for human health and agriculture counterbalance costs to meet Paris Agreement pledges. *Nat. Commun.* **9**, 4939 (2018).
24. Weyant, J. Some Contributions of Integrated Assessment Models of Global Climate Change. *Rev. Environ. Econ. Policy* **11**, 115–137 (2017).
25. Bullard, C. W. *Uncertainty in the 1967 US input-output data.* (CAC Document N^o 191. Center for Advanced Computation. University of Illinois at Urbana-Champaign, 1976).
26. Bullard, C. W. & Sebal, A. V. Monte Carlo Sensitivity Analysis of Input-Output Models. *Rev. Econ. Stat.* **70**, 708–712 (1988).
27. Dietzenbacher, E. The sensitivity of input-output multipliers. *J. Reg. Sci.* **30**, 239–258 (1990).
28. Dietzenbacher, E. & Los, B. Structural Decomposition Techniques: Sense and Sensitivity. *Econ. Syst. Res.* **10**, 307–324 (1998).
29. Hawkins, T., Hendrickson, C. & Matthews, H. S. Uncertainty in the mixed-unit input-output life cycle assessment (MUIO-LCA) model of the US Economy. *16th Int. Input-Output Conf. Int. Input-*

Output Assoc. (IIOA), 2-6 July 2007, Istanbul, Turkey (2007).

30. Kanemoto, K. & Tonooka, Y. Embodied CO₂ emissions in Japan's international trade. *J. Japan Soc. Energy Resour.* **30**, 15–23 (2009).
31. Lenzen, M. A generalized input-output multiplier calculus for Australia. *Econ. Syst. Res.* **13**, 65–92 (2001).
32. Lenzen, M., Pade, L. L. & Munksgaard, J. CO₂ multipliers in multi-region input-output models. *Econ. Syst. Res.* **16**, 389–412 (2004).
33. Lenzen, M. Aggregation versus disaggregation in input-output analysis of the environment. *Econ. Syst. Res.* **23**, 73–89 (2011).
34. Lenzen, M., Wood, R. & Wiedmann, T. Uncertainty Analysis for Multi-Region Input-Output Models – a Case Study of the UK'S Carbon Footprint. *Econ. Syst. Res.* **22**, 43–63 (2010).
35. Min, J. & Rao, N. D. Estimating Uncertainty in Household Energy Footprints. *J. Ind. Ecol.* **22**, 1307–1317 (2017).
36. Percoco, M., Hewings, G. J. D. & Senn, L. Structural change decomposition through a global sensitivity analysis of input-output models. *Econ. Syst. Res.* **18**, 115–131 (2006).
37. Peters, G. P. Opportunities and challenges for environmental MRIO modelling: Illustrations with the GTAP database. in *16th International Input-Output Conference of the International Input-Output Association (IIOA)* 1–26 (2007).
38. Roy, J. R. Regional input-output analysis, data and uncertainty. *Ann. Reg. Sci.* **38**, 397–412 (2004).
39. Tang, Z., Gong, P., Liu, W. & Li, J. Sensitivity of Chinese industrial wastewater discharge reduction to direct input coefficients in an input-output context. *Chinese Geogr. Sci.* **25**, 1–13 (2014).
40. Tarancón Morán, M. Á. & del Río González, P. A combined input-output and sensitivity analysis approach to analyse sector linkages and CO₂ emissions. *Energy Econ.* **29**, 578–597 (2007).
41. Temurshoev, U. Uncertainty Treatment in Input-Output Analysis. *Handb. Input-Output Anal.* 407–463 (2015). doi:10.4337/9781783476329.00018
42. Usubiaga, A. & Acosta-Fernández, J. Carbon emission accounting in MRIO models: the territory vs. the residence principle. *Econ. Syst. Res.* **27**, 458–477 (2015).
43. Weber, C. L., Matthews, H. S., Corbett, J. J. & Williams, E. D. Carbon emissions embodied in importation, transport and retail of electronics in the U.S.: a growing global issue. *IEEE Int. Symp. Electron. Environ. 7-10 May 2007* (2007).
44. Weber, C. L. Uncertainties in Constructing Environmental Multiregional Input-Output Models. *Int. Input Output Meet. Manag. Environ.* (2008).
45. Wiedmann, T., Lenzen, M. & Wood, R. *Uncertainty Analysis of the UK-MRIO Model—Results from a Monte-Carlo Analysis of the UK Multi-Region Input-Output Model (Embedded Carbon Dioxide Emissions Indicator); Report.* (2008).
46. Wiedmann, T., Wood, R., Minx, J., Lenzen, M. & Harris, R. Emissions embedded in UK trade - UK-MRIO model results and error estimates. *Int. Input-Output Meet. Manag. Environ. 9-11 July 2008, Seville, Spain* (2008).
47. Wilting, H. C. Sensitivity and uncertainty analysis in MRIO modelling; some empirical results with regard to the Dutch carbon footprint. *Econ. Syst. Res.* **24**, 141–171 (2012).
48. Benveniste, H., Boucher, O., Guivarch, C., Treut, H. Le & Criqui, P. Impacts of nationally determined contributions on 2030 global greenhouse gas emissions: uncertainty analysis and

distribution of emissions. *Environ. Res. Lett.* **13**, 14022 (2018).

Reviewers' comments:

Reviewer #1 (Remarks to the Author):

Dear authors,

Thank you for considering and reviewing in depth some of the issues highlighted in the list of publications I provided you, which could affect the implementation of your proposal. In my view, this more up-to-date and in-depth review of your proposal gives it much more value and places it, if possible, even more at the frontier of science.

For my part, as I have commented on in previous reviews, this article seems to me to be appropriate, essential and new enough to form part of a journal of the standing of Nature Communications. If we also consider the ability and willingness of the authors to understand, undertake and fit all the proposals made in the overall review process, I can only recommend this paper for publication.

Thank you for proposing a paper of such caliber and relevance.

Reviewer #2 (Remarks to the Author):

Dear authors

Thank you for improving the manuscript further. It's looking better and better. And the motivation for the non-expert has improved and that is important. I now "get" ERA much better from the beginning, the abstract and intro prepare the reader better for the technicalities later on. So far so good.

I have one remaining point. The authors write: "In 4 is rightfully stated that actions that contribute to reduced global emissions should be credited, and actions that increase them should be penalized. However, neither PBA nor CBA satisfy this principle." First, I think you need to clean up phrases like "In 4 is rightfully stated..." – the reference numbers don't work in the sentence flow. Now, let's look at the ability of actors to engage in actions that decrease emissions. Actors only have such capability (enabling them to receive ERA credits) if they have a choice, or other means of exerting power in the market. Eg if they are stuck with one product (eg in a monopolistic product market) then they have no chance of receiving ERA credits (because they cannot switch supplier, say if only Brazil produced meat). Or if they have limited product knowledge (eg consumers not seeing through some

technocratic marketing waffle or complicated product features, say if all cars came with near-incomprehensible technical manuals for petrol consumption features and control), then they are unable to decide which actions will get them ERAS credits. (These are only two examples; there may be more.) Here's where shared responsibility comes in: If the market is so monopolistic that it strips consumers of any power to bring about emission reductions, or if product information is so impenetrable that consumers cannot decide what actions will bring them ERA credits, then there is a rationale for shifting some of the responsibility to producers. In the end, in these situations (and others) they are the ones who have control over, and understand their product.

This is relevant in the context of this submission, because ERA adapts CBA. I see that the authors state "ERA adapts CBA. Most of the advantages and disadvantages of CBA hold therefore also for ERA. The advantages that are typically brought forward include: political benefits, more equity and justice 36,37,38, and providing a basis for border carbon taxes or adjustments 39. Disadvantages of CBA and implementations of the corresponding taxes and adjustments include concerns about effectiveness and efficiency, impediments of practical implementation, or political incompatibility (see 33)". Still, wouldn't it be straightforward to base ERA on shared responsibility accounting (SRA)? SRA shares all the mathematics with CBA, and so for example an alternative equation 2 could be written using the SRA notation in Gallego. The rest would have to be worked out.

With that, a part of the disadvantages of CBA that trickle into ERA would be dealt with, and the resulting ERA approach would be more general, fairer, and therefore more attractive. Please not I would not insist that the authors implement this generalisation in order to get published in NatComms, but I would like to ask that they consider this generalisation seriously, if not only because it might be so straightforward to implement it.

Comments on Reviewer #3 Remarks to the Author and authors' response:

Comment: - ... Despite all progress in input-output accounting, I do not share the perception that this COULD and SHOULD actually be used for climate policy-making due to the various sources of uncertainty involved (I provide more detailed explanations in my previous reviews - point 2). ...

Referee #2 comment: I believe that the existence of uncertainties for certain models or data does not preclude them to be used in decision-making. In fact, just about every facet of science is associated with uncertainty. The current problem is that decision-making is too often not taking uncertainty into consideration. Sound uncertainty estimates should i) be part of every good paper, and ii) be taken into account when making decisions.

Answer: ...

In our view "the various sources of uncertainty involved" are usually present in (basically) any model or approach for climate policy.

Referee #2 comment: I agree with this response.

In this regard, the question we pose is: can a system such as ERA, given the uncertainties it may have* (see below), be of use for climate policy? Or similarly, retaking an interesting question of the previous round of review made by Reviewer 3, do we expect that the proposed scheme could be the basis for certifiable and verifiable emission credits or penalties? Our answers are positive, based on our reasoning below.

Referee #2 comment: To dispel all doubts, the authors could actually perform a quantitative uncertainty analysis, using Monte-Carlo (MC) techniques. There are MRIO databases that come with accompanying standard deviation estimates, which in turn can be used to feed into an MC routine, which in turn is very easy to program. Then, the authors could actually demonstrate how error propagation – through the numerous additions underlying the matrix products involved in the ERA calculations – leads to aggregate measures (as opposed to sector- or region-specific multipliers and paths) being associated with surprisingly small errors. There are papers on error propagation for IO systems in the International Journal of Life-Cycle Assessment I believe. Having said this, I would not insist on MC analysis as a condition for publishing, but I do believe that such an addition would a) make this a very good paper and really support the “effective” claim in the title, and b) confirm once again a standard much needed: every (very) good paper should have results accompanied by quantitative uncertainty estimates. I guess this is an Editorial decision.

* We consider now Uncertainties for ERA:

- Input-output data gathering and modelling usual uncertainties: As discussed in works such as 25–47, and as pointed out in the previous round of reviews, error sources have been discussed extensively in the scientific literature. Based on the results from sensitivity analyses and knowing the key sources of uncertainty, researchers have evidence on the errors and their effects in specific cases and have been able to build up a good intuition.

Referee #2 comment: I agree with this response.

- Levels of regional/sectoral disaggregation: Among the sources of uncertainty we could cite, it was pointed out that aggregation bias could be a serious one, and that the ERA computations will be hard to perform at the level of ten-thousands of products that are internationally traded. We agree, real-world detail is not present in much of the information (e.g. on environmental pressures) provided to citizens and institutions. Also, we do not think that building global input-output databases at the level of ten-thousands of products can be expected in the near future.

Referee #2 comment: I agree with this response. However the issue already exists with 1,000 products. Making global MRIO tables is a highly underdetermined optimisation problem, with supporting constraints information typically being 100 times less numerous than MRIO table elements (eg 1000x1000 table with 1m elements compiled using 10,000 constraints). This really means that a large part of an MRIO table can be guesswork. But that may not matter actually. As Jensen (, R. C. The concept of accuracy in regional input-output models. International Regional Science Review 5,

139-154, 1980) has beautifully shown, one can delete a large part of small elements in a table and obtain virtually the same multipliers. So, the art here is to be aware of these issues, and compile and use the MRIO tables appropriately. This philosophy is for example at the heart of the new virtual MRIO labs, where users can build virtually any table, including ones with, say, a detailed disaggregation of regional hairdressing sectors. The problem is that users can build these tables without any raw data on regional hairdressing, and the MRIO info just comes from non-survey or other approaches to make the MRIO initial estimate. Of course, this would not make sense. But the good thing is, and this relates back to my earlier comment, virtual labs routinely offer quantitative info on standard deviations, and using this, one would quickly see that an analysis of regional hairdressing without raw data would lead to results being affected by significant uncertainties. Hence my recommendation above: use accompanying standard deviation info.

On the other hand, product information and labelling can be done at the detailed level

Referee #2 comment: This would be a problem if product-specific constraints were absent during MRIO compilation (and they very often are) – see my hairdressing example above. Plus, there are such large differences between product sub-types and brands (and these are important for labelling!), that labelling in my view is a long long way off.

and the evaluation at the aggregate level with input-output tables. For many products it suffices to know the emissions in the inputs and the emissions in the inputs into the inputs. We think that this information exactly incentivizes the behaviour of consumers that is aimed at in the ERA approach.

Referee #2 comment: I agree with this response.

- Choice of multiregional input-output database: Some researchers may argue that because of the existence of multiple accounting frameworks different answers will be provided to the same question. Agreement on the choice of the database (and for some questions also the method) may not occur. As we indicated before, e.g. Moran and Wood (2014) have shown that the main results for carbon footprint accounting converge across databases. If desired, one could average the results obtained from different input-output datasets. Furthermore, the more transparent these databases are, the easier it is to detect differences and underlying methodological choices, and ultimately arrive at convergence of results.

Referee #2 comment: I agree with this response. See 1 Owen, A., Steen-Olsen, K., Barrett, J., Wiedmann, T. & Lenzen, M. A structural decomposition approach to comparing MRIO databases. *Economic Systems Research* 26, 262-283, doi:10.1080/09535314.2014.935299 (2014).

Another development that we observe is that different institutions join forces and are moving towards a single global database. A major step forward is that a supranational institution like the OECD started to produce global MRIO tables. In this regard, there are interesting movements into the direction of generalizing accounting systems, notably regarding trade accounting (e.g. OECD and WTO joining forces in their TiVA initiatives¹).

Referee #2 comment: I agree with this response. But you could also mention the Réunion community as an example for people joining forces. This was a major development that led to the virtual MRIO labs.

Once the discrepancies between databases have reduced sufficiently, another executive decision could simply be: pick one database and require that everyone runs computations with it for a certain period of time. Resources may be spend on research to increase the number of countries and/or industries (for which researchers rely on solid national data, not using crude extrapolations).

Referee #2 comment: “Pick one database”. Not sure this works. If you had asked people in the Réunion project which one to pick, everyone would have picked their own. That was the problem. The solution is an MRIO database that hovers above all existing ones – the “Mother of all MRIOs” as one Réunion participant aptly coined the idea.

...

...

A final remark is that one does not need to provide all information at the most detailed level of products or industries. What is important is to signal the right incentives. For example, import energy intensive products from countries that produce them with less emitting technologies. The consequences will clearly be reflected in ERA. As highlighted by 17 for IAMs, we emphasize the need of transparency about the database and accuracy related to aggregation levels. This goes in line with the calls for a clearer framework for tracking contributions to reduce emissions.

Referee #2 comment: I agree with this response.

References of the Cover letter and Responses

1. OECD. Trade in value added. (2019). doi:<https://doi.org/10.1787/data-00648-en>
2. Fujimori, S., Hasegawa, T. & Masui, T. in (eds. Fujimori, S., Kainuma, M. & Masui, T.) 305–328 (Springer Singapore, 2017). doi:10.1007/978-981-10-3869-3_13
3. Edmonds, J. A. & Reily, J. M. User’s Guide to the IEA/ORAU Long-Term Global Energy Economic Model with Carbon Dioxide Emissions: Personal Computer Version A84PC. (1986).
4. Kim, S. H., Edmonds, J., Lurz, J., Smith, S. J. & Wise, M. The ObjECTS Framework for Integrated Assessment: Hybrid Modeling of Transportation. *Energy J. (Special Issue #2)* 51–80 (2006).
5. Stehfest, E. Integrated assessment of global environmental change with IMAGE 3.0 : model description and policy applications. (The Hague : PBL Netherlands Environmental Assessment Agency, 2014).

6. Krey, V. et al. MESSAGE-GLOBIOM 1.0 Documentation. (International Institute for Applied Systems Analysis (IIASA), 2016).
7. Fricko, O. et al. The marker quantification of the Shared Socioeconomic Pathway 2: A middle-of-the-road scenario for the 21st century. *Glob. Environ. Chang.* 42, 251–267 (2017).
8. Marian Leimbach Lavinia Baumstark, Michael Luken and Ottmar Edenhofer, N. B. Technological Change and International Trade - Insights from REMIND-R. *Energy J.* Volume 31, (2010).
9. Bosetti, V., Carraro, C., Galeotti, M., Massetti, E. & Tavoni, M. WITCH A World Induced Technical Change Hybrid Model. *Energy J.* 27, 13–37 (2006).
10. Manne, A., Mendelsohn, R. & Richels, R. MERGE: A model for evaluating regional and global effects of GHG reduction policies. *Energy Policy* 23, 17–34 (1995).
11. Blanford, G., Merrick, J., Richels, R. & Rose, S. Trade-offs between mitigation costs and temperature change. *Clim. Change* 123, 527–541 (2014).
12. Paltsev, S. et al. The MIT Emissions Predication and Policy Analysis (EPPA) Model: Version4. (2005).
13. Nordhaus, W. in *Handbook of Computable General Equilibrium Modeling SET*, Vols. 1A and 1B (eds. Dixon, P. B. & Jorgenson, D. W. B. T.-H. of C. G. E. M.) 1, 1069–1131 (Elsevier, 2013).
14. Dowlatabadi, H. Integrated assessment models of climate change: An incomplete overview. *Energy Policy* 23, 289–296 (1995).
15. Smith, S. J., Kim, S. H. & Pitcher, H. M. Model Documentation for the MiniCAM. (2003).
16. Wilkerson, J., Leibowicz, B., Diaz, D. & Weyant, J. Comparison of Integrated Assessment Models: Carbon Price Impacts on U.S. Energy. *Energy Policy* 76, (2015).
17. Schneider, S. H. Integrated assessment modeling of global climate change: Transparent rational tool for policy making or opaque screen hiding value-laden assumptions? *Environ. Model. Assess.* 2, 229–249 (1997).
18. Lawrence, M. G. et al. Evaluating climate geoengineering proposals in the context of the Paris Agreement temperature goals. *Nat. Commun.* 9, 3734 (2018).
19. Fletcher, S., Lickley, M. & Strzepek, K. Learning about climate change uncertainty enables flexible water infrastructure planning. *Nat. Commun.* 10, 1782 (2019).
20. Bordbar, M. H. et al. Uncertainty in near-term global surface warming linked to tropical Pacific climate variability. *Nat. Commun.* 10, 1990 (2019).
21. Yumashev, D. et al. Climate policy implications of nonlinear decline of Arctic land permafrost and other cryosphere elements. *Nat. Commun.* 10, 1900 (2019).
22. Scovronick, N. et al. The impact of human health co-benefits on evaluations of global climate policy. *Nat. Commun.* 10, 2095 (2019).

23. Vandyck, T. et al. Air quality co-benefits for human health and agriculture counterbalance costs to meet Paris Agreement pledges. *Nat. Commun.* 9, 4939 (2018).
24. Weyant, J. Some Contributions of Integrated Assessment Models of Global Climate Change. *Rev. Environ. Econ. Policy* 11, 115–137 (2017).
25. Bullard, C. W. Uncertainty in the 1967 US input-output data. (CAC Document No 191. Center for Advanced Computation. University of Illinois at Urbana-Champaign, 1976).
26. Bullard, C. W. & Sebald, A. V. Monte Carlo Sensitivity Analysis of Input-Output Models. *Rev. Econ. Stat.* 70, 708–712 (1988).
27. Dietzenbacher, E. The sensitivity of input-output multipliers. *J. Reg. Sci.* 30, 239–258 (1990).
28. Dietzenbacher, E. & Los, B. Structural Decomposition Techniques: Sense and Sensitivity. *Econ. Syst. Res.* 10, 307–324 (1998).
29. Hawkins, T., Hendrickson, C. & Matthews, H. S. Uncertainty in the mixed-unit input-output life cycle assessment (MUIO-LCA) model of the US Economy. 16th Int. Input-Output Conf. Int. Input-Output Assoc. (IIOA), 2-6 July 2007, Istanbul, Turkey (2007).
30. Kanemoto, K. & Tonooka, Y. Embodied CO₂ emissions in Japan's international trade. *J. Japan Soc. Energy Resour.* 30, 15–23 (2009).
31. Lenzen, M. A generalized input-output multiplier calculus for Australia. *Econ. Syst. Res.* 13, 65–92 (2001).
32. Lenzen, M., Pade, L. L. & Munksgaard, J. CO₂ multipliers in multi-region input-output models. *Econ. Syst. Res.* 16, 389–412 (2004).
33. Lenzen, M. Aggregation versus disaggregation in input-output analysis of the environment. *Econ. Syst. Res.* 23, 73–89 (2011).
34. Lenzen, M., Wood, R. & Wiedmann, T. Uncertainty Analysis for Multi-Region Input-Output Models – a Case Study of the UK'S Carbon Footprint. *Econ. Syst. Res.* 22, 43–63 (2010).
35. Min, J. & Rao, N. D. Estimating Uncertainty in Household Energy Footprints. *J. Ind. Ecol.* 22, 1307–1317 (2017).
36. Percoco, M., Hewings, G. J. D. & Senn, L. Structural change decomposition through a global sensitivity analysis of input-output models. *Econ. Syst. Res.* 18, 115–131 (2006).
37. Peters, G. P. Opportunities and challenges for environmental MRIO modelling: Illustrations with the GTAP database. in 16th International Input-Output Conference of the International Input-Output Association (IIOA) 1–26 (2007).
38. Roy, J. R. Regional input-output analysis, data and uncertainty. *Ann. Reg. Sci.* 38, 397–412 (2004).
39. Tang, Z., Gong, P., Liu, W. & Li, J. Sensitivity of Chinese industrial wastewater discharge reduction to direct input coefficients in an input-output context. *Chinese Geogr. Sci.* 25, 1–13 (2014).

40. Tarancón Morán, M. Á. & del Río González, P. A combined input-output and sensitivity analysis approach to analyse sector linkages and CO2 emissions. *Energy Econ.* 29, 578–597 (2007).
41. Temurshoev, U. Uncertainty Treatment in Input-Output Analysis. *Handb. Input–Output Anal.* 407–463 (2015). doi:10.4337/9781783476329.00018
42. Usubiaga, A. & Acosta-Fernández, J. Carbon emission accounting in MRIO models: the territory vs. the residence principle. *Econ. Syst. Res.* 27, 458–477 (2015).
43. Weber, C. L., Matthews, H. S., Corbett, J. J. & Williams, E. D. Carbon emissions embodied in importation, transport and retail of electronics in the U.S.: a growing global issue. *IEEE Int. Symp. Electron. Environ.* 7-10 May 2007 (2007).
44. Weber, C. L. Uncertainties in Constructing Environmental Multiregional Input-Output Models. *Int. Input Output Meet. Manag. Environ.* (2008).
45. Wiedmann, T., Lenzen, M. & Wood, R. Uncertainty Analysis of the UK-MRIO Model—Results from a Monte-Carlo Analysis of the UK Multi-Region Input–Output Model (Embedded Carbon Dioxide Emissions Indicator); Report. (2008).
46. Wiedmann, T., Wood, R., Minx, J., Lenzen, M. & Harris, R. Emissions embedded in UK trade - UK-MRIO model results and error estimates. *Int. Input-Output Meet. Manag. Environ.* 9-11 July 2008, Seville, Spain (2008).
47. Wilting, H. C. Sensitivity and uncertainty analysis in MRIO modelling; some empirical results with regard to the Dutch carbon footprint. *Econ. Syst. Res.* 24, 141–171 (2012).
48. Benveniste, H., Boucher, O., Guivarch, C., Treut, H. Le & Criqui, P. Impacts of nationally determined contributions on 2030 global greenhouse gas emissions: uncertainty analysis and distribution of emissions. *Environ. Res. Lett.* 13, 14022 (2018).

Reviewers' comments:

Reviewer #1 (Remarks to the Author):

Dear authors,

Thank you for considering and reviewing in depth some of the issues highlighted in the list of publications I provided you, which could affect the implementation of your proposal. In my view, this more up-to-date and in-depth review of your proposal gives it much more value and places it, if possible, even more at the frontier of science.

For my part, as I have commented on in previous reviews, this article seems to me to be appropriate, essential and new enough to form part of a journal of the standing of Nature Communications. If we also consider the ability and willingness of the authors to understand, undertake and fit all the proposals made in the overall review process, I can only recommend this paper for publication.

Thank you for proposing a paper of such caliber and relevance.

Answer: Thank you very much for the careful reviews.

Reviewer #2 (Remarks to the Author):

Dear authors

Thank you for improving the manuscript further. It's looking better and better. And the motivation for the non-expert has improved and that is important. I now "get" ERA much better from the beginning, the abstract and intro prepare the reader better for the technicalities later on. So far so good.

Thank you very much for the valuation, the in depth review and for even having discussed and proposed suggestions from the comments of the previous reviewer.

I have one remaining point. The authors write: "In 4 is rightfully stated that actions that contribute to reduced global emissions should be credited, and actions that increase them should be penalized. However, neither PBA nor CBA satisfy this principle." First, I think you need to clean up phrases like "In 4 is rightfully stated..." – the reference numbers don't work in the sentence flow. Now, let's look at the ability of actors to engage in actions that decrease emissions. Actors only have such capability (enabling them to receive ERA credits) if they have a choice, or other means of exerting power in the market. Eg if they are stuck with one product (eg in a monopolistic product market) then they have no chance of receiving ERA credits (because they cannot switch supplier, say if only Brazil produced meat). Or if they have limited product knowledge (eg consumers not seeing through some technocratic marketing waffle or complicated product features, say if all cars came with near-incomprehensible technical manuals for petrol consumption features and control), then they are unable to decide which actions will get them ERAS credits. (These are only two examples; there may be more.) Here's where shared responsibility comes in: If the market is so monopolistic that it strips consumers of any power to bring about emission reductions, or if product information is so impenetrable that consumers cannot decide what actions will bring them ERA credits, then there is a

rationale for shifting some of the responsibility to producers. In the end, in these situations (and others) they are the ones who have control over, and understand their product.

This is relevant in the context of this submission, because ERA adapts CBA. I see that the authors state “ERA adapts CBA. Most of the advantages and disadvantages of CBA hold therefore also for ERA. The advantages that are typically brought forward include: political benefits, more equity and justice 36,37,38, and providing a basis for border carbon taxes or adjustments 39. Disadvantages of CBA and implementations of the corresponding taxes and adjustments include concerns about effectiveness and efficiency, impediments of practical implementation, or political incompatibility (see 33)”. Still, wouldn’t it be straightforward to base ERA on shared responsibility accounting (SRA)? SRA shares all the mathematics with CBA, and so for example an alternative equation 2 could be written using the SRA notation in Gallego. The rest would have to be worked out.

With that, a part of the disadvantages of CBA that trickle into ERA would be dealt with, and the resulting ERA approach would be more general, fairer, and therefore more attractive. Please not I would not insist that the authors implement this generalisation in order to get published in NatComms, but I would like to ask that they consider this generalisation seriously, if not only because it might be so straightforward to implement it.

Answer: Thank you. We have added a new appendix to our Supplementary Information (Appendix D, the original Appendix D is now E). This new appendix:

- **mentions the caveat**
- **sketches how to develop a scheme of credits and penalties in the case of shared responsibility; and**
- **outlines such a scheme if we address producers directly and credit ‘better’ trade in intermediate products.**

The main text has been slightly adapted and mentions that (and briefly explains why) we focus on adapting CBA and on trade in final products.

Comments on Reviewer #3 Remarks to the Author and authors’ response:

Referee #2 comment: I believe that the existence of uncertainties for certain models or data does not preclude them to be used in decision-making. In fact, just about every facet of science is associated with uncertainty. The current problem is that decision-making is too often not taking uncertainty into consideration. Sound uncertainty estimates should i) be part of every good paper, and ii) be taken into account when making decisions.

Referee #2 comment: I agree with this response.

In this regard, the question we pose is: can a system such as ERA, given the uncertainties it may have (see below), be of use for climate policy? Or similarly, retaking an interesting question of the previous round of review made by Reviewer 3, do we expect that the proposed scheme could be the basis for certifiable and verifiable emission credits or penalties? Our answers are positive, based on our reasoning below.*

Referee #2 comment: To dispel all doubts, the authors could actually perform a quantitative uncertainty analysis, using Monte-Carlo (MC) techniques. There are MRIO databases that come with accompanying standard deviation estimates, which in turn can be used to feed into an MC routine, which in turn is very easy to program. Then, the authors could actually demonstrate how error propagation – through the numerous additions underlying the matrix products involved in the ERA calculations – leads to aggregate measures (as opposed to sector- or region-specific multipliers and paths) being associated with surprisingly small errors. There are

papers on error propagation for IO systems in the International Journal of Life-Cycle Assessment I believe. Having said this, I would not insist on MC analysis as a condition for publishing, but I do believe that such an addition would a) make this a very good paper and really support the “effective” claim in the title, and b) confirm once again a standard much needed: every (very) good paper should have results accompanied by quantitative uncertainty estimates. I guess this is an Editorial decision.

Answer: The last two comments and some below deal with uncertainty. We have carried out a quantitative analysis, using Monte-Carlo (MC) techniques in order to show uncertainty and suggest that decision-making also needs to consider it. We have discussed that there are MRIO databases that come with accompanying standard deviation estimates (see e.g.¹⁸) and that certainly it should be a common practice in MRIO analyses as well as on good empirical articles (this one has been moving through revisions from more of a theoretical one to having more and more empirical content).

** We consider now Uncertainties for ERA:*

- Input-output data gathering and modelling usual uncertainties: As discussed in works such as 25–47, and as pointed out in the previous round of reviews, error sources have been discussed extensively in the scientific literature. Based on the results from sensitivity analyses and knowing the key sources of uncertainty, researchers have evidence on the errors and their effects in specific cases and have been able to build up a good intuition. Referee #2 comment: I agree with this response.

- Levels of regional/sectoral disaggregation: Referee #2 comment: I agree with this response. However the issue already exists with 1,000 products. Making global MRIO tables is a highly underdetermined optimisation problem, with supporting constraints information typically being 100 times less numerous than MRIO table elements (eg 1000x1000 table with 1m elements compiled using 10,000 constraints). This really means that a large part of an MRIO table can be guesswork. But that may not matter actually. As Jensen (, R. C. The concept of accuracy in regional input-output models. International Regional Science Review 5, 139-154, 1980) has beautifully shown, one can delete a large part of small elements in a table and obtain virtually the same multipliers. So, the art here is to be aware of these issues, and compile and use the MRIO tables appropriately. This philosophy is for example at the heart of the new virtual MRIO labs, where users can build virtually any table, including ones with, say, a detailed disaggregation of regional hairdressing sectors. The problem is that users can build these tables without any raw data on regional hairdressing, and the MRIO info just comes from non-survey or other approaches to make the MRIO initial estimate. Of course, this would not make sense. But the good thing is, and this relates back to my earlier comment, virtual labs routinely offer quantitative info on standard deviations, and using this, one would quickly see that an analysis of regional hairdressing without raw data would lead to results being affected by significant uncertainties. Hence my recommendation above: use accompanying standard deviation info.

Answer: The paper by Jensen is part of a research tradition that was very popular in the 1970s and 1980s. It covered topics such as Fundamental Economic Structure and Inverse Important Coefficients. As indicated by the reviewer virtual labs —like many things in life— can be very wonderful when used appropriately but ‘harmful’ when abused or used inappropriately. We are not sure whether standard deviations solve the problem (although they contain very useful information and ought to be included more often). One of the lessons from the research on Inverse Important Coefficients is that a small error in one input coefficient can have a much larger effect than large errors in some other coefficients. This information might be used to reduce the uncertainty in these inverse important coefficients.

We have briefly cited and discussed some of the literature (related to input-output) on these matters of uncertainty analyses. In the Monte Carlo analysis we have briefly introduced this discussion and accompanied standard deviations as well as additional analyses (e.g. simply on percentage changes obtained from different runs) to reveal which type of dimensions and variables are affected more significantly to uncertainties and potential errors.

On the other hand, product information and labelling can be done at the detailed level

Referee #2 comment: This would be a problem if product-specific constraints were absent during MRIO compilation (and they very often are) – see my hairdressing example above. Plus, there are such large differences between product sub-types and brands (and these are important for labelling!), that labelling in my view is a long long way off.

Answer: We see the point made by the referee. We have noticed the problem of product-specific constraints in MRIO compilation. We also seem to be more optimistic in some respects. For example, to promote that consumers use more healthy products, governments are developing regulations to make nutritional information easier accessible by requiring so-called front-of-package labels (FOPLs). Given the speed of digitalization, such FOPLs might be within reach in the not too distant future also for emissions. In any case, these aspects have been briefly mentioned in the manuscript and supplementary material.

(...)and the evaluation at the aggregate level with input-output tables. For many products it suffices to know the emissions in the inputs and the emissions in the inputs into the inputs. We think that this information exactly incentivizes the behaviour of consumers that is aimed at in the ERA approach. Referee #2 comment: I agree with this response.

Choice of multiregional input-output database: Some researchers may argue that because of the existence of multiple accounting frameworks different answers will be provided to the same question. Agreement on the choice of the database (and for some questions also the method) may not occur. As we indicated before, e.g. Moran and Wood (2014) have shown that the main results for carbon footprint accounting converge across databases. If desired, one could average the results obtained from different input-output datasets. Furthermore, the more transparent these databases are, the easier it is to detect differences and underlying methodological choices, and ultimately arrive at convergence of results.

Referee #2 comment: I agree with this response. See 1 Owen, A., Steen-Olsen, K., Barrett, J., Wiedmann, T. & Lenzen, M. A structural decomposition approach to comparing MRIO databases. Economic Systems Research 26, 262-283, doi:10.1080/09535314.2014.935299 (2014).

Answer: We have added the reference to this paper.

Another development that we observe is that different institutions join forces and are moving towards a single global database. A major step forward is that a supranational institution like the OECD started to produce global MRIO tables. In this regard, there are interesting movements into the direction of generalizing accounting systems, notably regarding trade accounting (e.g. OECD and WTO joining forces in their TiVA initiatives¹).

Referee #2 comment: I agree with this response. But you could also mention the Réunion community as an example for people joining forces. This was a major development that led to the virtual MRIO labs.

Answer: We have added extra text. “Another example is the Project Réunion in which a group of researchers (representing the existing GMRIO datasets) met to discuss the way forward, which was an important incentive for forming the virtual MRIO labs (see the

reflections on the Project Réunion in Lenzen et al, 2017, see also Tukker and Dietzenbacher, 2013).”

Once the discrepancies between databases have reduced sufficiently, another executive decision could simply be: pick one database and require that everyone runs computations with it for a certain period of time. Resources may be spend on research to increase the number of countries and/or industries (for which researchers rely on solid national data, not using crude extrapolations).

Referee #2 comment: “Pick one database”. Not sure this works. If you had asked people in the Réunion project which one to pick, everyone would have picked their own. That was the problem. The solution is an MRIO database that hovers above all existing ones – the “Mother of all MRIOs” as one Réunion participant aptly coined the idea.

Answer: The same participant added that it would be “a huge mama”. This is also why probably the participants to this project should not be the ones to ask to pick. Moreover, one of the conclusions from the project (and summarized in the ESR paper by Tukker and Dietzenbacher, 2013) is that one dataset is more suitable to answer one type of questions whereas another dataset should be used for another set of questions.

The ideal GMRIO table does not exist, so it would have been a valid option (in our view) if a supranational organization taking care of global emissions and climate change had picked one of the datasets. Because the “Mother” does not exist, we will always make errors, no matter which GMRIO is chosen. At a certain stage, to have a standard becomes very important in our opinion.

A final remark is that one does not need to provide all information at the most detailed level of products or industries. What is important is to signal the right incentives. For example, import energy intensive products from countries that produce them with less emitting technologies. The consequences will clearly be reflected in ERA. As highlighted by 17 for IAMs, we emphasize the need of transparency about the database and accuracy related to aggregation levels. This goes in line with the calls for a clearer framework for tracking contributions to reduce emissions.

Referee #2 comment: I agree with this response.

We are happy to be in agreement with several responses. Again, a great thank you for the review comments and for having considered and proposed suggestions from the comments of the previous reviewer as well.

REVIEWERS' COMMENTS:

Reviewer #2 (Remarks to the Author):

Dear authors

I've studied the revisions and responses to my earlier comments. Well done, I have no further concerns. I do hope though that our work in this area will lead to some concrete change in the world, sooner rather than later. The bushfire crisis in Australia (or better called the on-set of a full-blown climate crisis?) has given us a stark reminder of what's in store. As such, I hope that you may join the push for the UNFCCC to adopt some measure of consumption-based accounting. In this sense, seeing people work on making such accounting schemes fairer and more acceptable gives me hope.